# ON REDUNDANCY AND DIVERSITY IN CELL-BASED NEURAL ARCHITECTURE SEARCH

**Xingchen Wan**[1,§]**, Binxin Ru**[1,§]**, Pedro M. Esperança**[2]**, Zhenguo Li**[3]
[1]Machine Learning Research Group, University of Oxford
[2]Huawei Noah's Ark Lab, London [3]Huawei Noah's Ark Lab, Hong Kong
`{xwan,robin}@robots.ox.ac.uk, {pedro.esperanca,li.zhenguo}@huawei.com`

## ABSTRACT

Searching for the architecture cells is a dominant paradigm in NAS. However, little attention has been devoted to the analysis of the cell-based search spaces even though it is highly important for the continual development of NAS. In this work, we conduct an empirical post-hoc analysis of architectures from the popular cell-based search spaces and find that the existing search spaces contain a high degree of redundancy: the architecture performance is minimally sensitive to changes at large parts of the cells, and universally adopted designs, like the explicit search for a reduction cell, significantly increase the complexities but have very limited impact on the performance. Across architectures found by a diverse set of search strategies, we consistently find that the parts of the cells that do matter for architecture performance often follow similar and simple patterns. By explicitly constraining cells to include these patterns, randomly sampled architectures can match or even outperform the state of the art. These findings cast doubts into our ability to discover truly novel architectures in the existing cell-based search spaces, and inspire our suggestions for improvement to guide future NAS research. Code is available at https://github.com/xingchenwan/cell-based-NAS-analysis.

## 1 INTRODUCTION

Neural Architecture Search (NAS), which automates designs of neural networks by task, has seen enormous advancements since its invention. In particular, *cell-based* NAS has become an important technique in NAS research: in contrast to attempts that directly aim to design architectures at once and inspired by the classical, manually-designed architectures like VGGNet and ResNet that feature repeated blocks, cell-based NAS searches for repeated *cells* only and later stacks them into full architectures. This simplification reduces the search space (although it remains highly complex), and allows easy transfer of architectures across different tasks/datasets, application scenarios and resources constraints (Elsken et al., 2019). Indeed, although alternative search spaces exist, cell-based NAS has received an extraordinary amount of research attention: based on our preliminary survey, **almost 80%** of the papers proposing new NAS methods published in ICLR, ICML and NEURIPS in the past year show at least one part of their major results in standard Differentiable Architecture Search (DARTS) cell-based spaces and/or highly related ones; and **approx. 60%** demonstrate results in such spaces *only* (detailed in App C). It is fair to state that such cell-based spaces currently dominates.

However, the lagging understanding of these architectures and the search space itself stands in stark contrast with the volume and pace of new *search algorithms*. The literature on *understanding why this dominant search space works* and *comparing what different* NAS *methods have found* is much more limited in depth and scope, with most existing works typically focusing exclusively on a few search methods or highlighting high-level patterns only (Shu et al., 2020; Zela et al., 2020). We argue that strengthening such an understanding is crucial on multiple fronts, and the lack thereof is concerning: first, studying a diverse set of architectures enables us to discover patterns shared across the search space, the foundation of comparison amongst search methods that heavily influences the resultant performances (Yang et al., 2020a): if the search space itself is flawed, designing and iterating new search methods on it, which is typically computationally intensive, can be misleading and unproductive. Conversely, understanding any pitfalls of the existing search spaces informs us on how to design better search spaces in the future, which is critical in advancing the goal of NAS in

---

§Work done while interning at Huawei Noah's Ark Lab, London, UK.

finding novel and high-performing architectures, not only in conventional CNNs but also in emerging NAS paradigms such as Transformers (which may also take the form of a cell-based design space). Second, opening the NAS black box enables us to distill the essence of the strong-performing NAS architectures beneath their surface of complexity. Unlike manually designed architectures where usually designers attribute performance to specific designs, currently owing to the apparent complexity of the design space, the NAS architectures, while all discovered in a similar or identical search space, are often compared to in terms of final performance on a standard dataset only (e.g. CIFAR-10 test error). This could be problematic, as we do not necessarily understand *what* NAS has discovered that led to the purported improvements, and the metric itself is a poor one on which external factors such as hyperparameter settings, variations in training/data augmentation protocols and even just noise could exert a greater influence than the architectural design itself (Yang et al., 2020a). However, by linking performance to specific designs, we could ascertain whether any performance differences stem from the architectures rather than the interfering factors.

We aim to address this problem by presenting a post-hoc analysis of the well-performing architectures produced by technically diverse search methods. Specifically, we utilise explainable machine learning tools to open the NAS black box by inspecting the good- and bad-performing architectures produced by a wide range of NAS search methods in the dominant DARTS search space. We find:

- Performances of architectures can often be disproportionately attributed to a small number of simple yet critical features that resemble *known* patterns in classical network designs;
- Many designs almost universally adopted contribute to complexity but not performance;
- The nominal complexity of the search spaces poorly reflects the actual diversity of the (high-performing) architectures discovered, the functional parts of which are often very similar despite the technical diversity in search methods and the seeming disparity in topology.

In fact, with few simple and human-interpretable constraints, almost *any* randomly sampled architecture can perform on par or exceed those produced by the state-of-the-art NAS methods over varying network sizes and datasets (CIFAR-10/IMAGENET). Ultimately, these findings prompt us to rethink the suitability of the current standard protocol in evaluating NAS and the capability to find truly novel architectures within the search space. We finally provide suggestions for prospective new search spaces inspired by these findings.

## 2 PRELIMINARIES

DARTS space (Fig 1) proposed by Liu et al. (2019)—which is in turn inspired by Zoph et al. (2018) and Pham et al. (2018)—is the most influential search space in cell-based NAS: it takes a form of a Directed Acyclic Graph (DAG), which features 2 inputs (connected to the output of two immediately preceding cells), 1 output and 4 intermediate nodes. The *operation* $o^{(i,j)}$ on the $(i,j)$-th edge is selected from a set of candidate *primitives* $\mathbb{A}$ with size $K = 7$ to transform $x^{(i)}$. At each intermediate node, the output from the operations on all its predecessors are aggregated: $x^{(j)} = \sum_{i<j} o^{(i,j)}(x^{(i)})$. And finally the out

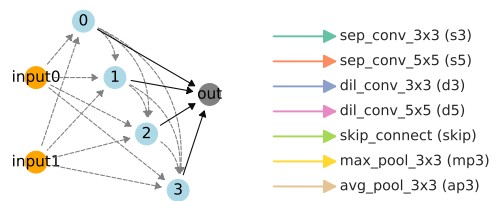

Figure 1: The DARTS cell. The solid black arrows denote the concatenation at output which is fixed. The gray dashed arrows denote the 14 potential operation locations. In a valid DARTS cell, 8 of which are filled by one out of the seven candidate primitives listed to the right, while the other 6 spots are disabled.

node concatenates all outputs from the intermediate nodes. As shown in Fig 1, the DARTS cell allows for up to 14 edges/operations. However, to ensure that all intermediate nodes are connected in the computational graph, each intermediate node is constrained to have exactly 2 in-edges connected to it from 2 different preceding nodes. Lastly, it is conventional to search for two cells, namely *normal* $\alpha_n$ and *reduce* $\alpha_r$ cells simultaneously, with $\alpha_r$ placed at 1/3 and 2/3 of the total depth of the networks and $\alpha_n$ everywhere else. In total, there are $\prod_{k=1}^{4} \frac{7^2 k(k+1)}{2} \approx 10^9$ distinct cells without considering graph isomorphism, and since both $\alpha_n$ and $\alpha_r$ are required to fully specify an architecture, there exist approx. $(10^9)^2 = 10^{18}$ distinct architectures (Liu et al., 2019). Other cells commonly used are almost invariably closely related to, and can be viewed as simplified or more complicated variants of the DARTS space. For example, the NAS-Bench-201 (NB201) (Dong & Yang, 2020) search space is simplified from the DARTS cell (detailed and analysed in App D). Other works have increased the number of intermediate nodes (Wang et al., 2021b), expanded the primitive pool $\mathbb{A}$ (Hundt et al., 2019) and/or relaxed other constraints (e.g. Shi et al. (2020) allow both incoming edges of the

intermediate nodes to be from the same preceding nodes), but do not fundamentally differ from the DARTS space. Additionally, search spaces like the NAS-Bench-101 (NB101) (Ying et al., 2019a) feature cells where the operations are represented as node features, although we believe that the way the cells are represented should not affect any findings we will demonstrate, as the DARTS/NB201 spaces can similarly be equivalently represented in a feature-on-node style (Ru et al., 2021; Pham et al., 2018). We do not experiment on NB101 in the present paper, as it features CIFAR-10 only, which is similar to NAS-Bench-301 (NB301) (Siems et al., 2020), but the latter is much closer in terms of size to a "realistic" search space that the community expects and is more amenable to one-shot methods which are currently mainstream in NAS.

## 3 OPERATION-LEVEL ANALYSIS: REDUNDANCIES IN SEARCH SPACES

The cell-based NAS search space contains multiple sources of complexity: deciding both the specific wiring and operations on the edges; searching for 2 cells independently, etc.; each expanding the search space combinatorially. While this complexity is argued to be necessary for the search space to be expressive for good-performing, novel architectures to be found, it is unknown whether *all* these sources of complexities are equally important, or whether the performance critically depend on some sub-features only. We argue that answering this question, and identifying such features, if they are indeed present, is highly significant: it enables us to separate the complexities that positively contribute to performances from those that do not, which could fundamentally affect the design of search methods. At the level of individual architectures, it helps us understand what NAS has truly discovered, by removing confounding factors and focusing on the key aspects of the architectures.

For findings to be generally applicable, we aim not to be specific to any search method; this requires us to study a large set of architectures that are high-performing but technically diverse in terms of the search methods that produce them. Fortunately, the training set of NB301 (Siems et al., 2020)—which includes 50,000+ architecture–performance pairs in the DARTS space using a combination of random sampling and more than 10 state-of-the-art yet technically diverse methods (detailed in App. B.1)—could be used for such an analysis: to build a surrogate model that accurately predicts architecture performance across the entire space. We primarily focus on the top-5% (2,589) architectures of the training set since we overwhelmingly care about the good-performing architectures by definition in NAS, although as we will show, the main findings hold true also for architectures found by methods not covered by NB301 and for other search spaces like the NB201 space. As shown in Fig 2, the top architectures are well-spread out in the search space and are well-separated by search methods, seemingly suggesting that the architectures discovered by different methods are diverse in characteristics. Lastly, the worse-performing cells could also be of interest, as any features observed could be the ones we would actively like to avoid, and we analyse them in App. A.

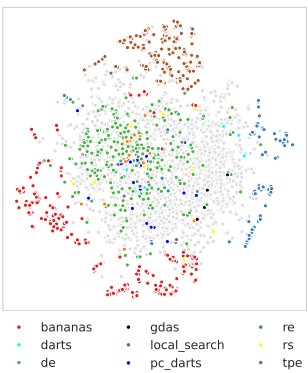

bananas    gdas    re
darts    local_search    rs
de    pc_darts    tpe
drnas    random

Figure 2: *The top archs provide a good coverage of the search space and are seemingly diverse*: the t-SNE plot of the top 5% archs (colored markers; grouped by different search methods. Details in App B.1) and randomly sampled archs in the DARTS search space (gray markers).

**Operation Importance** To untangle the influence of each part of the architecture, we introduce *Operation Importance (*OI*)*, which measures the incremental effect of the individual operations (the smallest possible features) to the overall performance. We quantify this via measuring the expected change in the performance by perturbing the type or wiring of the operation in question: considering an edge-attributed cell $\alpha$ with edge $e_{i,j}$ currently assigned with primitive $o_k$, then the operation importance of $o_k$ on that edge of cell $\alpha$ is given by:

$$\text{OI}(\alpha, e_{i,j} := o_k) = \frac{1}{|\mathcal{N}(\alpha, e_{i,j} := o_k)|} \sum_{m=1}^{|\mathcal{N}(\alpha, e_{i,j} := o_k)|} \Big[ y(\alpha_m) \Big] - y(\alpha), \tag{1}$$

where $y(\alpha)$ denotes the validation accuracy or another appropriate performance metric of the fully-trained architecture induced by cell $\alpha$. We are interested in measuring the importance of both the primitive choice and *where the operation is located relative to the entire cell*, and we use $\mathcal{N}(\alpha, e_{i,j} = o_k)$ to denote a set of neighbour cells that differ to $\alpha$ only with the edge in question assigned with another primitive, $e_{i,j} \in \mathbb{A} \setminus \{o_k\}$; or with the same primitive $o_k$ but with one end

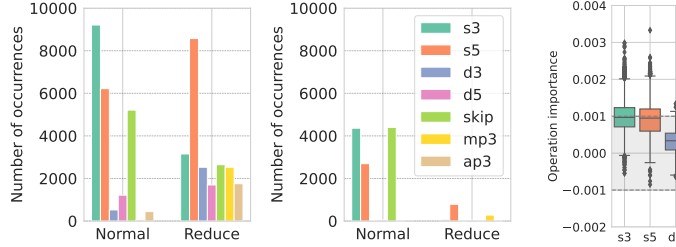

(a) *All* operations      (b) *Important* operations

Figure 3: Distribution of (a) all and (b) important operations by the primitive types of the top-performing archiectures, organised by primitive type.

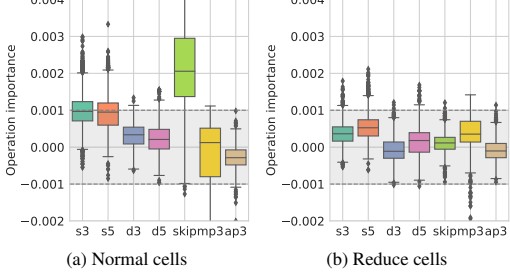

(a) Normal cells      (b) Reduce cells

Figure 4: OI distribution in (a) normal and (b) reduce cells. The important operations are shown outside the gray shaded area.

node of $e_{i,j}$ being rewired to another node, subjected to any constraints in the search space. It is worth noting that OI is an instance of the *Permutation Feature Importance (*PFI*)* (Breiman, 2001; Fisher et al., 2019; Molnar, 2019). Given the categorical nature of the "features" in this case, we may enumerate all permutations on the edge in question instead of having to rely on random sampling as conventional PFI does. An important operation by Eq (1) would therefore attributed an OI of large magnitude in either direction, whereas an irrelevant one would have a value of zero since altering it on expectation leads to no change in architecture performance. We compute the OI of each operation for the 2,589 architectures. To circumvent the computational challenge of having to train all neighbour architectures of $\alpha$ (which are outside the NB301 training set) from scratch to compute $y(\cdot)$, we use the performance prediction $\tilde{y}(\cdot)$ from NB301. However, as we will show, we validate all key findings by actually training some architectures to ensure that they are not artefacts of the statistical surrogate (training protocols detailed in App. B.2).

**Findings**    The most natural way to group the operations is by their primitive and cell (i.e. normal or reduce) types, and we show the main results in Figs. 3 and 4. In Fig. 3(b), we discretise the OI scores using the threshold 0.001 (0.1%)—which is similar to the observed noise standard deviation of the better-performing architectures from NB301— to highlight the *important operations* with $|\text{OI}| \geq 0.001$: these are the ones we could more confidently assert to affect the architecture performance beyond noise effects. We summarise the key findings below:

*(#1) Only a fraction of operations is critical for good performance within cells:*   If all operations need to be fully specified for good performance (i.e., with both the primitive choice and the specific wiring determined), then perturbing any of them should lead to significant performance deterioration. However, this is clearly not the case in practice: comparing Fig. 3(a) and (b), we observe that only a fraction of the operations are important based on our definition. To verify this directly beyond predicted performances, we randomly select 30 architectures from the top 5% training set. Within each cell, we sort their 16 operations by their OI in both ascending and descending orders. We then successively disable the operations by zeroing them, and train the resulting architectures with increasing number of operations disabled from scratch[1] until only half of the active operations remain (Fig. 5). The results largely confirms our findings and shows that the OI, although computed via predicted performance, is accurate in representing the ground-truth importance of the operations. On average, we need to disable 6 low-OI operations

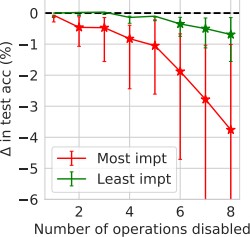

Figure 5: Ground-truth change in accuracy by successively disabling the most/least important ops, ordered by their OI. Medians and interquartile ranges shown; stars denote that the drop in accuracy is significant at $p \leq 0.01$ in the Wilcoxon signed-rank test.

to see a statistically significant drop in performance, and almost half of the operations to match the effect of disabling just 2 high-OI ones. On the other hand, disabling the high-OI operations quickly reduce the performance and in some cases stall the training altogether. Noting that the overall standard deviation of the NB301 training set is just 0.8%, the drop in performance is quite dramatic.

*(#2) Reduce cells are relatively unimportant for good performance:*   Searching independently for reduce cells scale the search space quadratically, but Fig. 3(b) shows that they contain much fewer important operations, and Fig. 4 shows that the OI distribution across all primitives are centered close to zero in reduce cells: the reduce cell is therefore less important to the architecture performance. To verify this, we draw another 30 random architectures. For each of the them, we construct and train from scratch the original architecture and 4 derived ones, with (a) reduce cell set identical to normal

---

[1]Note that we may not obtain NB301 performance prediction on these architectures, as NB301 requires all 16 operations to be enabled with valid primitives.

cell, (b) reduce cell with all operations set to parameterless skip connections, (c) normal cell set identical to reduce cell and (d) normal cell with operations set to skip connections. Setting the reduce cell to be identical to the normal cell leads to *no significant change in performance*, while the reverse is not true (Fig. 6). A more extreme example is that while setting cells to consist of skip connections only is unsurprisingly sub-optimal in both cases, doing so on the reduce cells harms the performance much less. This suggests that while searching separately for reduce cells are well-motivated, the current design, which places much fewer reduce cells than normal cells in the overall architecture yet treats them equally during search, might be a sub-optimal trade-off; and searching two separate cells may yield little benefits over the strategy of simply using the same searching rule and applying it on both normal and reduce cells.

*(#3) Different primitives have vastly different importance profiles with many of them redundant:* The set of candidate primitives $\mathbb{A}$ consists of operators that are known to be useful in manually-designed architectures, with the expectation that they should also be, to varying degrees, useful to NAS. However, this is clearly not the case: while it is already known that some primitives are favoured more by certain search algorithms (Zela et al., 2020), the observed discrepancy in the relative importance of the primitives is, in fact, more extreme: the normal cells (which are also the important cells by Finding #2) across the entire spectrum of good performing architectures overwhelmingly favour only 3 out of 7 possible primitives: separable convolutions and skip connection (Fig. 3(a)). Even when the remaining 4 primitives are occasionally selected, they are almost never important (Fig. 3(b)). This is also observed in Fig. 4(a) which shows that they have distributions of OI close to 0. As we will show later in Sec. 4, we could essentially remove these primitives from $\mathbb{A}$ without impacting the performances. Even within the 3 favoured primitives, there is a significant amount of variation. First, comparing Figs 1(a) and (b), `skips`, when present in good architectures, are very likely to be important. We also note that the distribution of OI of `skip` has a higher variance – these suggest that the performance of an architecture is highly sensitive towards the specific locations and patterns of skip connections, a detailed study of which we defer to

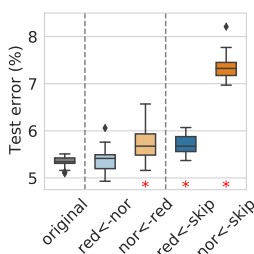

Figure 6: Ground-truth test errors (i.e. not predicted by NB301) of the original archs (`original`), archs with reduce cells set identical to their normal cells (`red<-nor`)/normal cells set identical to their reduce cells (`nor<-red`) and archs with normal/reduce cells fully replaced by skip connections (`nor<-skip`/`red<-skip`). ∗ denotes that the performance distribution significantly differs from `original` at $p \leq 0.01$ in the Wilcoxon signed-rank test.

Sec. 4. On the other hand, while both separable convolutions (`s3` and `s5`) are highly over-represented in the good-performing architectures, it seems that they are less important than `skip`. A possible explanation is that while their presence is required for good performance, their exact locations in the cell matter less, which we again verify in Sec. 4. Increasingly we are interested in multi-objectives (e.g. maximising performance *while* minimising costs); we show that even accounting for this, redundant primitives, especially pooling, remain largely redundant (App. G).

**Discussions** The findings confirm that both the search space and the cells contain various redundancies that increase the search complexity but do not actually contribute much to performance; and that good performance most often does not depend on an entire cell but a few key features and primitives in both individual cells and in the search space. This clearly shows that the search space design can be further optimised, but consequently, many beliefs often ingrained in existing search methods can also be sub-optimal or unnecessary. For example, barring a few exceptions (Xie et al., 2019a; You et al., 2020; Ru et al., 2020; Wan et al., 2022), the overwhelming majority of the current approaches aims to search for a single, fully deterministic architecture, and this often results in high-dimensional vector encoding of the cells (e.g. the path encoding of DARTS cell in White et al. (2021) is $> 10^4$ dimensions without truncation). This affects the performance in general (White et al., 2020a) and impedes methods that suffer from curse of dimensionality, such as Gaussian Processes, in particular. However, exact encoding could be in fact unnecessary if good performance simply hinges upon only a few key designs while the rest does not matter as much, and finding relevant low-dimensional, *approximate* representations could be beneficial instead.

## 4 SUBGRAPH-LEVEL ANALYSIS: ARE WE TRULY FINDING NOVEL CELLS?

Sec. 3 demonstrates the *presence* of critical sub-features within good performing architectures. In this section we aim to find what they actually are and whether there are commonalities amongst the

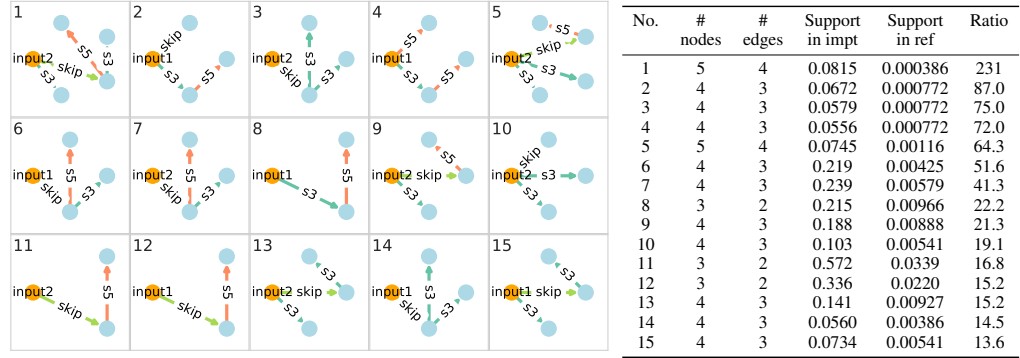

| No. | # nodes | # edges | Support in impt | Support in ref | Ratio |
|-----|---------|---------|-----------------|----------------|-------|
| 1 | 5 | 4 | 0.0815 | 0.000386 | 231 |
| 2 | 4 | 3 | 0.0672 | 0.000772 | 87.0 |
| 3 | 4 | 3 | 0.0579 | 0.000772 | 75.0 |
| 4 | 4 | 3 | 0.0556 | 0.000772 | 72.0 |
| 5 | 5 | 4 | 0.0745 | 0.00116 | 64.3 |
| 6 | 4 | 3 | 0.219 | 0.00425 | 51.6 |
| 7 | 4 | 3 | 0.239 | 0.00579 | 41.3 |
| 8 | 3 | 2 | 0.215 | 0.00966 | 22.2 |
| 9 | 4 | 3 | 0.188 | 0.00888 | 21.3 |
| 10 | 4 | 3 | 0.103 | 0.00541 | 19.1 |
| 11 | 3 | 2 | 0.572 | 0.0339 | 16.8 |
| 12 | 3 | 2 | 0.336 | 0.0220 | 15.2 |
| 13 | 4 | 3 | 0.141 | 0.00927 | 15.2 |
| 14 | 4 | 3 | 0.0560 | 0.00386 | 14.5 |
| 15 | 4 | 3 | 0.0734 | 0.00541 | 13.6 |

Figure 7 & Table 1: Frequent subgraphs in the good-performing architectures ranked by ratio of supports and their properties. Almost all subgraphs feature `skip` residual links with additional connections with 1 or more `sep_convs` and neither `dil_convs` nor other parameterless operations.

architectures found by technically diverse methods. Towards this goal, operation-level analysis is insufficient as the performance of neural networks also depends on the architecture topology and graph properties of the wiring between the operations (Xie et al., 2019a; Ru et al., 2020).

**Frequent Subgraph Mining (FSM)** FSM aims to "to extract all the frequent subgraphs, in a given data set, whose occurrence counts are above a specified threshold" (Jiang et al., 2013). This is immensely useful for our use-case, as any frequent subgraphs mined on the architectures represented by DAGs would naturally represent the interesting recurring structures in the good-performing architectures, and subgraphs are also widely used for generating explanations (Ying et al., 2019b). In our specific case, we 1) convert the computing graphs corresponding to the topology of the same set of top-performing architectures in Sec. 3 into DAGs, 2) within each DAG, we retain only the important operations defined in Sec 3 and 3) run an adapted version of the gSpan algorithm (Yan & Han, 2002) for DAGs on the set of all architecture cell graphs $\{G_1, ..., G_T\}$ to identify a set of the most frequent subgraphs $\mathcal{G}^f = \{g_1^f, ..., g_M^f\}$ where each subgraph must have a minimum *support* $S_{(.)}$ of $\sigma = 0.05$:

$$S_{g_i^f} = \frac{|\delta(g_i^f)|}{T} \geq \sigma \, \forall g_i^f \in \mathcal{G}^f, \text{ where } \delta(g_i^f) = \{G_j | g_i^f \subseteq G_j\}_{j=1}^T. \quad (2)$$

One caveat with *support* is that it favours simpler subgraphs, which are more likely to be present "by nature". To account for this bias, we measure the significance of these subgraphs over *a null reference* using the *ratio* between supports of $\mathcal{G}^f$ and the reference. The readers are referred to App. I for detailed explanations.

**Findings** *(#4) Functional parts of many good-performing architectures are structurally similar to each other and to elements of classical architectures.* We show the top subgraphs in terms of the ratio of supports in Fig. 7, and an immediate insight is that the top frequent subgraphs representing the common traits across the entire good-performing region of the search space are highly over-represented over the reference and *non-diverse*: almost all subgraphs can be characterised with skip connections forming residual links between one or both input nodes with an operation node, combined with different number and/or sizes of separable convolutions. In fact, we find that this ResNet-style residual link is present in 98.5% (2,815) of the top 2,859 architectures (as a comparison, if we sample randomly, only approximately half of the architectures are expected to contain this feature). With reference to Fig. 8, the residual links

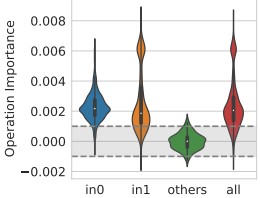

Figure 8: `skips` are only use-ful when they form residual links: `in0` and `in1` denote the residual links formed with either inputs, `others` denote the skip connections not forming residual links and `all` is the overall distribution of OI of skip connections.

*drive* the importance of `skip` in Fig. 3. This suggests that skip connections do not just benefit optimisation of NAS supernets but also actively contribute to generalisation if they posit as residual connections. The propensity of certain NAS methods of collapsing into cells almost entirely consisting of `skip` is well-known with many possible explanations and remedies, but here we provide an alternative perspective independent of search methods: more fundamentally, `skip` is the only primitive whose exact position greatly impacts the performances in both positive and negative directions and thus it is more difficult for search methods learn such a relation precisely.

The consensus in preferring the aforementioned pattern also extends beyond the training set of NB301: with reference to Fig. 9, we select some of the architectures produced by the most recent works that are not represented in the NB301 training set, and it is clear that despite the different

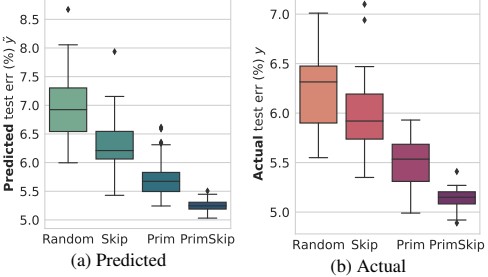

(a) White et al. (2021)  (b) Chen et al. (2021b)  (c) Li et al. (2021)  (d) Ru et al. (2021)  (e) Wang et al. (2021c)  (f) Chen et al. (2021a)

Figure 9: Normal cells of various SoTA (left to right: BANANAS, DRNAS, GAEA, NAS-BOWL, DARTS_PT and TE-NAS) architectures with the important operations highlighted (the connections to output are omitted since they are all identical across the DARTS search space). Note all cases considered are consistent with the residual link + separable convolution patterns identified, even though the cells and search methods are very different and except for BANANAS, none of the methods here was used to generate the NB301 training set.

search strategies, functional parts of resulting architectures are all characterised by the this pattern of residual connections and separable convolutions, a combination already well-known and well-used both in parts and in sum in successful manually designed networks (e.g. Xception (Chollet, 2017) uses both ingradients). In this sense, many of the existing NAS methods might not have discovered much more novel architectures beyond what we already know; the functional parts of many SoTA NAS architectures could be regarded as variations of the classical architectures, whereas the apparent diversity like the one shown in Fig. 2 is often caused by differences in the non-functional parts of the architectures that in fact minimally influence the performances.

***Generating* SoTA architectures**  We showed that many NAS architectures share similar traits, but a stronger test is whether these simple patterns alone are *sufficient* for good performance. We construct architectures simply by random sampling, but with 2 constraints, *without additional search*:

1. Normals cell must contain residual link: for architecture generation, we simply manually wire 2 `skip`s from both inputs to intermediate node 0 (*Skip* constraint);
2. The other operations are selected from {`s3`, `s5`} only, with all other primitives removed (*Prim* constraint).

While it takes a thorough analysis to study the patterns, the constraints which encode our findings themselves are simple, human-interpretable and moderate (note that only *Skip* is a "hard" constraint specifying exact wiring; *Prim* simply constrains sampling to a smaller subspace). We then sample 100 architectures within both constraints with the same rule for both normal and reduce cells and report their predicted test errors in Fig 10(a). To ensure that we are not biased by the NB301 surrogate, we actually train 30 of the 100 architectures from scratch (protocols specified in App. B.2) and report results in Fig 10(b). To verify the relative importance of each constraint, we also sample the same number of architectures with no constraints or with either constraint activated. We note that both constraints effectively narrow the spread of both predicted and actual test errors, with almost *any* architecture in the *PrimSkip* group performing similarly to the SoTA – only <1% of the training set of NB301 perform better than the mean predicted test error of the *PrimSkip* group (5.24%) in Fig 10(a), while only 5% perform better than the *worst* (5.46%). Apart from the two constraints, the architectures produced are rather varied otherwise in terms of depth/width and exact wiring (see App. E) – this shows that the two moderate constraints already determine the performance to a great extent, potentially eclipsing other factors previously believed to influence performance. We believe that it might even be possible to fully construct architectures manually from the identified patterns to achieve better results, but the main purpose of this experiment is to show that we may narrowly constrain performance to a very competitive range using very few rules without exactly specifying the cells, instead of aiming for the absolute best architecture in an already noisy search space. Lastly, we analyse NB201 space similarly in App. D and very similar findings hold.

Figure 10: Distribution of (a) NB301 predicted and (b) actual test error of archs sampled. *Random*: random archs without constraints; *Skip*: archs with residual links and otherwise randomly sampled; *Prim*: random archs using {`s3`, `s5`, `skip`} only. *PrimSkip*: archs satisfying both *Skip* and *Prim*.

**Large architectures**  We have so far followed the standard NB301 training protocol featuring smaller architectures trained with fewer (100) epochs, generalisation performance on which does not necessarily always transfer to larger architectures (Yang et al., 2020a; Shu et al., 2020). Since the computational cost is much larger, here we first evaluate on 2 *PrimSkip* architectures from Sec. 4 with different width/depth profiles, but stack it into a larger architectures and train longer to make the results comparable those reported the literature. To ensure that the results are *completely* comparable,

Table 2: Test error of the state-of-the-art architectures on CIFAR-10 and IMAGENET (mobile setting).

(a) CIFAR-10. All baselines are re-evaluated using the procedure in App. B.2 to ensure the results are completely comparable.

| Architecture | Top-1 test error (%) | | Edit dist. |
| | Original | Edited | |
| --- | --- | --- | --- |
| DARTSv2 (Liu et al., 2019) | 2.44 | 2.36(−0.08) | 1 |
| BANANAS (White et al., 2021) | 2.39 | 2.42(+0.03) | 1 |
| DrNAS (Chen et al., 2021b) | **2.27** | 2.31(+0.04) | 1 |
| GAEA (Li et al., 2021) | 2.31 | **2.18**(−0.13) | 0 |
| NAS-BOWL (Ru et al., 2021) | 2.33 | 2.23(−0.10) | 2 |
| NoisyDARTS (Chu et al., 2020) | 2.57 | 2.42(−0.15) | 4 |
| DARTS_PT (Wang et al., 2021c) | 2.33 | 2.35(+0.02) | 2 |
| SDARTS_PT (Wang et al., 2021c) | 2.46 | 2.36(−0.10) | 4 |
| SGAS_PT (Wang et al., 2021c) | 2.92 | 2.48(−0.44) | 3 |
| *PrimSkip Arch 1* | **2.27** | - | - |
| *PrimSkip Arch 2* | 2.29 | - | - |

(b) ImageNet. All baselines are taken from the original papers as re-evaluation is too costly in this case.

| Architecture | Test error (%) | | Params (M) |
| | Top-1 | Top-5 | |
| --- | --- | --- | --- |
| DARTSv2 (Liu et al., 2019) | 26.7 | 8.7 | 4.7 |
| SNAS (Xie et al., 2019b) | 27.3 | 9.2 | 4.3 |
| GDAS (Dong & Yang, 2020) | 26.0 | 8.5 | 5.3 |
| DrNAS[†] (Chen et al., 2021b) | 24.2 | 7.3 | 5.2 |
| GAEA(C10) (Li et al., 2021) | 24.3 | 7.3 | 5.3 |
| GAEA(ImageNet)[†] (Li et al., 2021) | 24.0 | 7.3 | 5.6 |
| PDARTS (Chen et al., 2019) | 24.4 | 7.4 | 4.9 |
| PC-DARTS(C10) (Xu et al., 2020) | 25.1 | 7.8 | 5.3 |
| PC-DARTS(ImageNet)[†] (Xu et al., 2020) | 24.2 | 7.3 | 5.3 |
| *PrimSkip Arch 1* | 24.4 | 7.4 | 5.7 |
| *PrimSkip Arch 2* | **23.9** | **7.0** | 5.7 |

[†]: searched directly on ImageNet.

on CIFAR-10 experiments we do not simply take the baseline results from the original papers; instead, we obtain the cell specifications provided in some of the most recent papers (see App. E for detailed specifications), re-train each from scratch using a standardised setup (See App. B.3 for details), and show the performance in the "*Original*" column of Table 2a. Recognising that the performance on CIFAR-10 is usually quite noisy (std dev around $0.05 - 0.1\%$), the sampled architectures perform at least on par with the SoTA architectures produced from much more sophisticated search algorithms.

To further verify our findings, we conduct an additional experiment where we edit the SoTA architectures minimally to make them comply to the *PrimSkip* constraints: whenever a cell contains primitives outside {s3, s5, skip}, we replace them with ones that are in this set, and we always set the reduce cell to be identical to the normal cell (see App. E for detailed specifications of the architectures). We also create residual links if they are not present, and replace non-residual skips with s3/s5 between operations, if any; we do not alter any wiring. Most architectures are already close to conforming to the constraints, so the number of edits required is often small (edit distances are under "*Edit dist.*" column in Table 2a); in fact, the GAEA cell is already fully-compliant and we only replace its reduce cell with the normal cell). We show the test errors of the edited architectures along with the change from the original ones in "*Edited*" column of Table 2a: the edits result in an improvement in test error up to 0.44% in 6/9 cases, and even where test errors increase after the edits, the differences are marginal and probably within margins of error. This shows that at least for the architectures we consider, the SoTA architectures can all be consistently explained by the same simple pattern identified. We finally train the same sampled *PrimSkip* architectures on ImageNet using a training protocol adopted in the literature (App. B.3) and the results are shown in Table 2b. Accounting for the estimated evaluation noise (most NAS papers do not run ImageNet experiments with multiple seeds, but comparable works like Xie et al. (2019a); Goyal et al. (2017) estimate a noise standard deviation of $0.2 - 0.5\%$ in Top-1 error), it is fair to say that both *PrimSkip* architectures perform on par to, if not better than, the SoTA, even though some of the SoTA architectures are searched on ImageNet directly, an extremely expensive procedure in terms of computational costs.

**Discussions** We find that the despite the different search methods and the belief that the search space contains diverse solutions, the key features of many top architectures in the DARTS (and NB201) space are largely similar to each other, and may collectively be viewed as variants to classical architectures. This suggests a gap between the apparent and effective diversity of the search space, potentially explaining the discrepancy between the huge search space and the small performance variability. We also show highly complicated SoTA methods fail to significantly outperform random search with few mild constraints, further demonstrating that it is these simple traits, and not other complexities, that drive the performance. Consequently, we argue that we should rethink the role that existing cell-based search spaces play as the key (and sometimes the only) venue on which the search methods develop and iterate. While it is reassuring that we find elements known to perform well, we should not over-rely on such search spaces but should continuously improve on them. Also, while we do not rule out the possibility that there could be other good-performing architectures not represented by the patterns identified, it is doubtful whether the search space, while nominally huge, truly contains novel and performant architectures beyond our knowledge.

## 5 RELATED WORKS

There are multiple previous works that also aim to explain and/or find patterns in cell-based NAS: Shu et al. (2020) find search methods in the DARTS space to favour shallow and wide cells but conclude they do not necessarily generalise better, and hence the pattern does not *explain* performances. Ru et al. (2021) also use subgraphs to explain performances, but only consider first-order Weisfeiler-Lehman

(WL) features which could be overly restrictive (note that most subgraphs we find in Fig. 7 are not limited to 1-WL features) and specific to the search method proposed. Zela et al. (2020) account for failure mode of differentiable NAS, but ultimately focus on a family of related search methods while the current work is search method-agnostic. On a search space level, a closely related work is Yang et al. (2020a), findings from whom we use extensively, but they mainly identify problems, not explanations; Xie et al. (2019a), Ru et al. (2020) and You et al. (2020) relate performances with graph theoretic properties of networks, but it is unclear to what extent these apply to standard cell-based NAS as the search spaces considered are significantly different (e.g. they typically feature much fewer primitive choices). Lastly, in constructing NAS benchmarks, Dong & Yang (2020); Siems et al. (2020); Ying et al. (2019a) have also provided various insights and patterns, but current work advances such understanding further via a comprehensive and experimentally validated investigation.

## 6 Suggestions for Future NAS Practices

We believe this work to be useful as an investigation of the existing cell-based NAS as well as to inspire future ones, not only on conventional CNNs but also emerging architectures like Transformers. On a search space level, we find a mismatch between the nominal and the effective complexity: complexities are as useful as they contribute to performance and novelty, and thus in a hypothetical new space, we should aim to be aware of these non-functional complexities, and not simply augment the number of primitives available and/or expanding the sizes of the cells. However, identifying such redundancies in a new search space is very challenging a-priori, but fortunately the analysis tools used in this paper are model-agnostic, and thus could be applied to any new search space candidates. Also, while we use the NB301 predictors which train a huge number of architectures to ensure the findings are as representative as possible, we show in App. F that combined with an appropriate surrogate regression model, we may reproduce most findings by training as few as 200 architectures (or 0.4% of the full training set); this suggests that the techniques used could also be cost-effective tools to inspect new search spaces. Another under-explored possibility would be iterative search-and-prune at the search space level, as opposed to the architecture level which is relatively well studied. Using the tools and metrics we introduced to incrementally *grow* the search space from simpler structures and *prune* out those redundant ones in a principled manner.

Notwithstanding the issues identified, we believe that the cell-based design paradigm remains valuable for proper benchmarking and comparison of NAS methods, but the many woes could be due to the over-engineered cells and the under-engineered macro-connections often fixed in a way that is heavily borrowed from classical networks. Here we identify three promising directions that would hopefully resolve or at least alleviate some of the problems identified: Firstly, we could simplify the cells but relax the constraints on how different cells are connected in the search space. Secondly, as discussed, we might be implicitly biased to favour architectures that resemble known architectures, preventing NAS to discover truly novel architectures. A possible solution is to *search on a more fundamental level* free of the bias of existing designs such as the paradigm championed by Real et al. (2020). Lastly, considering how much benchmarks have democratised and popularised NAS research, there is also a need for NAS benchmarks beyond simple cell-based spaces. The readers are referred to App. J for a more detailed discussions on our suggestions for future NAS researchers.

## 7 Conclusion

We present a post-hoc analysis of architectures in the most popular cell-based search spaces. We find a mismatch between the nominal the effective complexity, as many good-performing architectures, despite discovered by very different search methods, share similar traits. We also find many redundant design options, as performances of the architectures disproportionately depend on certain patterns while the rest are often irrelevant. We conclude that like the rapidly iterating search methods, the search spaces also need to evolve to match the progress of NAS and we provide suggestions based on the main findings in the paper, the latter of which also form some of the most evident directions of future work. Lastly, while cell-based NAS focusing on image classification is currently mainstream and the present paper focuses exclusively on such, alternative spaces and/or tasks exist (Howard et al., 2019; Wu et al., 2019; Cai et al., 2019; Duan et al., 2021; Tu et al., 2021). By adopting a macro search framework or focusing on an alternative task, they might be less subjected to some of the issues identified. Since the tools presented in the paper are largely search space-agnostic, a further future direction would also be to extend some of the analyses to them.

## ETHICS STATEMENT

While we do not see immediate ethical repercussions of our work in particular, we believe that the general topics of neural architecture search and automated machine learning (AutoML) that our work focuses on do involve broader interests at stake. On the positive side, gaining more knowledge on AutoML and NAS could lead to improved democratisation of deep learning models to non-experts as they automate machine learning pipelines that previously could require immense human expertise. On the negative side, it is worth noting that NAS and AutoML (including some of our suggestions for future directions in Sec. 6) are often computationally expensive. Advancement in AutoML and NAS might prompt more practitioners to use them instead of off-the-shelf models, which could lead to increased monetary and environmental costs. Finally, while AutoML and NAS themselves are ethically neural, there is possibility for them to be misused in ethically dubious ways. We believe that ultimately the practitioners and researchers have to make the judgement call to ensure that ethical use of the technologies in their domain of application is always ensured.

## REPRODUCIBILITY STATEMENT

The implementations details to reproduce the major experimental results of the paper are included in App. B. The code is open-sourced at https://github.com/xingchenwan/cell-based-NAS-analysis.

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

# APPENDICES

## A    ANALYSIS ON WORST-PERFORMING ARCHITECTURES

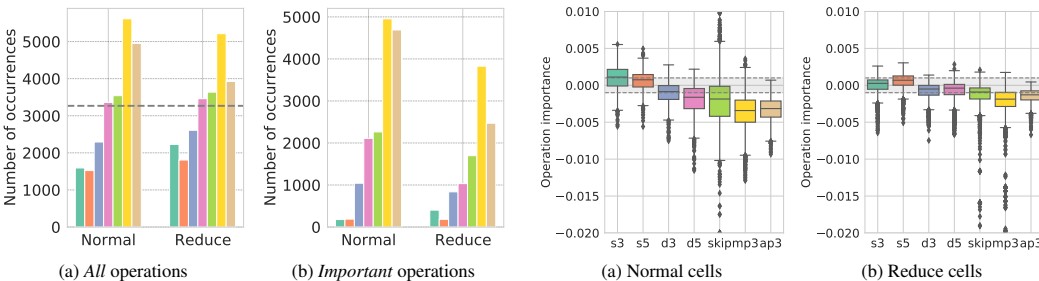

| (a) *All* operations | (b) *Important* operations | (a) Normal cells | (b) Reduce cells |

Figure 11: Distribution of (a) all and (b) important operations by the primitive type of the worst-performing cells. The gray dashed line in (a) denotes the expected number of occurrences if the operations are uniformly sampled in each cell.

Figure 12: Box-and-whisker plots showing the distribution of the operations importance in (a) normal and (b) reduce cells by primitive types. The important operations by the definition of the paper are shown outside the gray shaded area.

We also conduct operation-level analysis on the worst 5% performing architectures of the NB301 training data and the results are shown in Figs 11 and 12, and we find that both pooling operators almost never contribute to good performing architectures as shown in Sec 3, but they actively hurt performance in the poor architectures: from Fig 11, it is clear that the worst performing cells are both characterised by large number of pooling operators (Fig 11(a) and a large number of *important* pooling operators actively degrading performance: this is unsurprising and also pointed out in the analysis in the original NB301 paper (Siems et al., 2020) as cells with a large number of pooling operations aggressively cause loss of information. Other than that, both dilated convolution operations remain rather neutral and the separable convolutions remain positive in operation importance even in poorly-performing architectures. Skip connections in this case are quite negative in general but still have a very large spread – this could be either due to a large number of skip connections in the cell which is a known failure mode of many differentiable NAS algorithms Zela et al. (2020) or that skip connections need to be paired with separable convolutions as shown in Fig 7 for positive effects, which is not the case in these poor architectures where separable convolutions are underrepresented.

We also repeat the subgraph-level analysis on these architectures (Fig 13). An interesting insight is that the poorly performing subgraphs are much more "diverse" than the good ones, and the primitives in the two groups almost never overlap: the none of positive subgraphs in 7 contains {d3, d5, mp3, ap3}, whereas none of the negative subgraphs contains {s3, s5}. This shows that in the present search space the primitives are somewhat "separable" and the redundant primitives {d3, d5, mp3, ap3} may simply be discarded without affecting the resulting performances – we argue this should not be the case in a well-designed ideal search space. In principle, every primitive should be a building block that that potentially contribute to architectures positively at least in some cases.

## B    IMPLEMENTATION DETAILS

### B.1    DATA

We primarily use the data from the training set of the NB301 benchmark (Siems et al., 2020), available at the official repository at https://github.com/automl/nasbench301. To obtain a sound performance surrogate over the entire DARTS search space, NB301 trains more than 50,000 architectures using the protocol listed in App B.2, and use the architectures and their corresponding test performance on CIFAR-10 as inputs and labels to train a number of surrogate models as performance predictors, including GIN, XGBoost and LGBoost (in this paper, we always use the XGBoost surrogate as it is shown to be the best on balance according to Siems et al. (2020)). The architectures are produced from a number of technically diverse methods representing almost all mainstream genres of cell-based NAS like gradient-based methods, Bayesian optimisation, reinforcement learning and simpler methods like local search and random search: DARTS (Liu et al., 2019), DRNAS (Chen et al., 2021b), GDAS (Dong & Yang, 2020), reinforcement learning (RL) (Zoph & Le, 2016), differential evolution (DE) (Price et al., 2006), PC_DARTS (Xu et al., 2020), Tree parzen

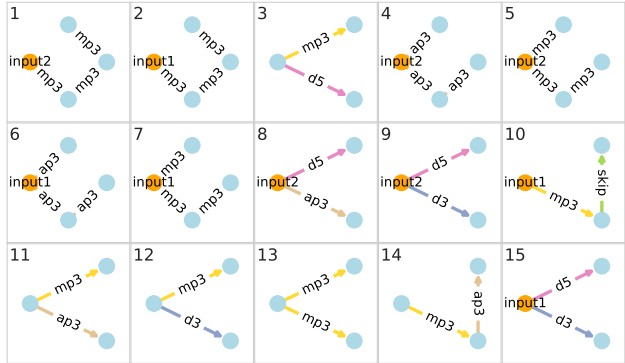

Figure 13: Frequent subgraphs in the good-performing architectures ranked by ratio of supports between the important subgraphs and the reference and properties of the discovered frequent subgraphs in the worst-performing architectures.

estimator (TPE) (Bergstra et al., 2011), local search (LS) (Den Ottelander et al., 2021; White et al., 2020b) and regularised evolution (RE) (Real et al., 2019).

## B.2 TRAINING PROTOCOLS ON DARTS ARCHITECTURES

We exactly follow the NB301 protocols for all experiments involving architecture training (except for the larger architectures on CIFAR-10 and ImageNet, which we outline below at App B.3). Specifically, we train architectures obtained from stacking the cells 8 times (8-layer architectures) with initial channel count of 32 on the CIFAR-10 dataset using the standard train/val split, and we use the hyperparameters below on a single NVIDIA Tesla V100 GPU:

```
Optimizer: SGD
Initial learning rate: 0.025
Final learning rate: 1e-8
Learning rate schedule: cosine annealing
Epochs: 100
Weight decay: 3e-4
Momentum: 0.9
Auxiliary tower: True
Auxliary weight: 0.4
Cutout: True
Cutout length: 16
Drop path probability: 0.2
Gradient clip: 5
Batch size: 96
Mixup: True
Mixup alpha: 0.2
```

## B.3 EVALUATION PROTOCOLS ON DARTS ARCHITECTURES

**CIFAR-10** For the evaluation, we use larger architectures obtained from stacking the cells 20 times (20-layer architectures) with an initial channel count of 36 on the CIFAR-10 dataset. The other hyperparameters (mostly consistent with those used in App B.2 except for the number of epochs trained) used are:

```
Optimizer: SGD
Initial learning rate: 0.025
Final learning rate: 1e-8
Learning rate schedule: cosine annealing
Epochs: 600
Weight decay: 3e-4
Momentum: 0.9
Auxiliary tower: True
Auxliary weight: 0.4
```

```
Cutout: True
Cutout length: 16
Drop path probability: 0.2
Gradient clip: 5
Batch size: 96
Mixup: True
Mixup alpha: 0.2
```

This protocol is identical to the original DARTS protocol (Liu et al., 2019), with the only exception that to be consistent with the NB301 protocol, we also incorporate the Mixup regularisation (Zhang et al., 2017) during evaluation. This accounts for the fact that the accuracy reported in this paper is generally better than those reported in the literature. However, as mentioned in the main text, we re-train every architectures from the scratch, including the baselines, using the identical protocol listed above, instead of simply taking the numbers from the original papers. As a result, no architecture has been given unfair advantage because of the more effective regularisation used in this paper. We also conduct all experiments on a single NVIDIA Tesla V100 GPU.

**ImageNet**   On ImageNet, we use a protocol that is identical to Chen et al. (2021b). It is also almost identical to those used in Xu et al. (2020); Chen et al. (2019); Liu et al. (2019) except for batch sizes (which depend on the availability of hardware; larger batch size is only available for a parallel many-GPU setup) and the corresponding linear scaling in learning rates. Specifically, we form 14-layer architectures with an initial channel count of 48 using $8\times$ NVIDIA Tesla V100 GPUs. Note that since we are unable to re-evaluate all the baselines using a standardised training protocols in this case due to the extreme computational cost, we use a protocol that strictly adheres to the existing works with the additional Mixup regularisation in CIFAR-10 disabled in the ImageNet experiments to ensure the comparability of the results. The other hyperparameters are as followed:

```
Optimizer: SGD
Initial learning rate: 0.5
Learning rate schedule: linear annealing
Epochs: 250
Weight decay: 3e-5
Momentum: 0.9
Auxiliary Tower: True
Auxliary weight: 0.4
Cutout: True
Cutout length: 16
Drop path probability: 0
Gradient clip: 5
Label smoothing: 0.1
Mixup: False
Batch size: 768
```

## C   LIST OF REFERENCED PAPERS

We present the details covered in our preliminary survey on the NAS search methods papers published in top machine learning conferences during the past year in Table 3.

Table 3: A list of NAS methods papers (i.e. excluding, e.g. review or benchmark papers) published in the past year in top machine learning conferences. *Cells-based* means the work demonstrates at least one part of the major results in the DARTS cell-based search space and/or highly related ones (such as the various NAS-Benches and/or those otherwise highly resemble DARTS). *Cells-only* means the works *only* demonstrate the results in aforementioned search space(s). Whenever a paper is not cell-based or cells-only, *other spaces evaluated* shows the alternative spaces the papers report results on. The list is potentially incomplete, as we only select papers that explicitly mention NAS in the title and/or the abstract.

| Venue | Name | Reference | Cells-based | Cells-only | Other spaces evaluated |
|---|---|---|---|---|---|
| NeurIPS 2020 | BRP-NAS | Dudziak et al. (2020) | ✓ | ✓ | |
| | NAGO | Ru et al. (2020) | | | NAGO space |
| | ISTA-NAS | Yang et al. (2020b) | ✓ | ✓ | |
| | arch2vec | Yan et al. (2020) | ✓ | ✓ | |
| | - | White et al. (2020a) | ✓ | ✓ | |
| | PR-DARTS | Zhou et al. (2020) | ✓ | ✓ | |
| | E²NAS | Zhang et al. (2020) | ✓ | ✓ | |
| | APS | Wang et al. (2020) | | | Channel/width search |
| | SemiNAS | Luo et al. (2020) | ✓ | | MobileNet space |
| | BONAS | Shi et al. (2020) | ✓ | ✓ | |
| ICLR 2021 | NAS-BOWL | Ru et al. (2021) | ✓ | ✓ | |
| | DrNAS | Chen et al. (2021b) | ✓ | ✓ | |
| | GAEA | Li et al. (2021) | ✓ | ✓ | |
| | DARTS- | Chu et al. (2021) | ✓ | ✓ | |
| | TE-NAS | Chen et al. (2021a) | ✓ | ✓ | |
| | DARTS_PT, etc | Wang et al. (2021c) | ✓ | | MobileNet space |
| | MetaD2A | Lee et al. (2021) | ✓ | | MobileNet space |
| | CafeNet | Su et al. (2021a) | | | Channel/width search |
| ICML 2021 | BO-TW/kDPP | Nguyen et al. (2021) | ✓ | ✓ | |
| | AlphaNet | Wang et al. (2021a) | | | MobileNet space |
| | CATE | Wang et al. (2021a) | ✓ | ✓ | |
| | HardCoRe-NAS | Nayman et al. (2021) | | | MobileNet space |
| | K-Shot NAS | Su et al. (2021b) | ✓ | | MobileNet space |
| | Few-shot NAS | Zhao et al. (2021) | ✓ | | ProxylessNAS space, RNN, AutoGAN |
| **Total** | **24** | | **19** (79%) | **14** (58 %) | |

## D  ANALYSIS ON NAS-BENCH-201

Fig 14 shows the NB201 search space, a popular NAS benchmark commonly used that is highly similar to the DARTS cell, but 1) only one cell (instead of two) is searched, 2) each cell is connected to its immediate preceding layer only, and 3) is with a reduced set of primitives. Also, unlike the DARTS cell, all edges in the NB201 cell are enabled.

We also conduct a brief analysis in a similar manner to the main text on top 5% performing architectures on NB201 dataset, and we show the operation importance distribution of each primitive in Fig 15. We observe that due to the smaller cell size and the primitive set, the operations in a NB201 cell is typically more

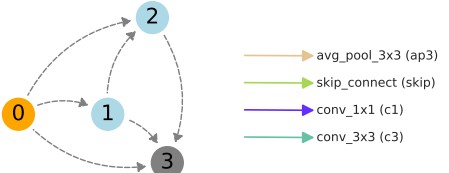

Figure 14: The NB201 (Dong & Yang, 2020) cell, which is highly similar to the DARTS space but much simpler. All 6 locations (denoted by gray dashed arrows) are available for search, and each is filled by one out of the four candidate primitives (or None, which disables the edge).

important and the only redundant operations is ap3. We hypothesise that the reason is similar to the DARTS search space as the manually specified macro connection between the cells already include pooling operations, rendering them unncessary within the cells.

The second experiment to conduct is verifying whether in the NB201 search space the good performing cells are also characterised by the patterns we identified in Sec 4. To do so, we adapt the *Skip* and *Prim* constraints in the NB201 space:

1. *Skip* constraint: in the NB201 search space, the only way to form a residual connection is to place skip on edge $0 \rightarrow 3$ (with reference to Fig 14.

2. *Prim* constraint: apart from the manually specified edge, all other operations are sampled from the reduced primitive set {c1, c3} consisting of convolutions only.

Similar to our procedure in Sec 4, we sample 50 architectures within each group (no constraint, either constraint and both constraints), and we show their test performance in Fig 16. It is also worth noting

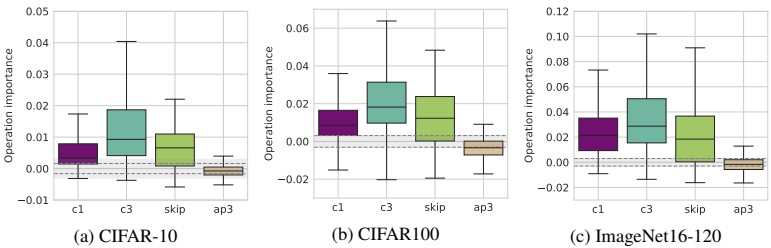

Figure 15: Box-and-whisker plots showing the distribution of the operation distribution in NB201 benchmark on (a) CIFAR-10, (b) CIFAR-100 and (c) ImageNet16-120 datasets. The gray shaded areas denote the noise standard deviation which differs in each dataset.

that the ground-truth optimum in each dataset is known in NB201 and is accordingly marked in Fig 16. Differing from the observations in DARTS search space results, in this case *Skip* constraint alone does not impact the performance significantly, but again the *PrimSkip* group with both constraints activated perform in a range very close to the optimum: in fact, the optimal architectures in all 3 datasets, while different from each other, all belong to the *PrimSkip* group and are found by random sampling with fewer than 50 samples. This again confirms that our findings in the main text similarly generalise to NB201 space.

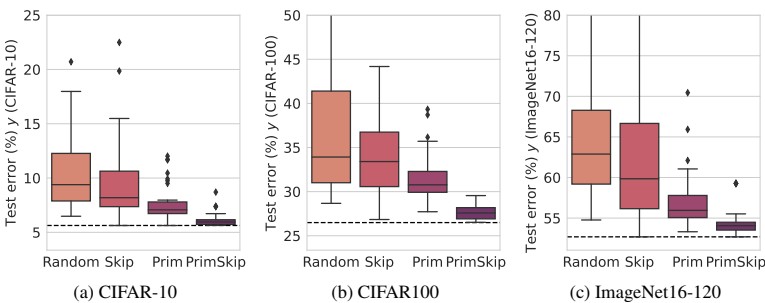

Figure 16: Distribution of the test errors on (a) CIFAR-10, (b) CIFAR-100 and (c) ImageNet of NB201 architectures. Note that since NB201 is a tabular benchmark that exhaustively trains and evaluates all the architectures within its search space, all test errors reported here are actual, not predicted.

# E  ARCHITECTURE SPECIFICATIONS

In this section, we show the specifications of (a.k.a genotypes) the different architectures in the DARTS search space.

## E.1  ORIGINAL AND EDITED GENOTYPES FROM BASELINE PAPERS

Here we show the genotypes original and edited (corresponding to the results in the *"Edited"* column in Table 2a) architectures (Fig. 17 – 24). In all figures, "Normal" and "Reduce" denote the normal and reduce cells of the *Original* architectures where "Edited" denote the normal and reduce cells of the edited architectures (note that the edited architectures always have identical normal and reduce cells).

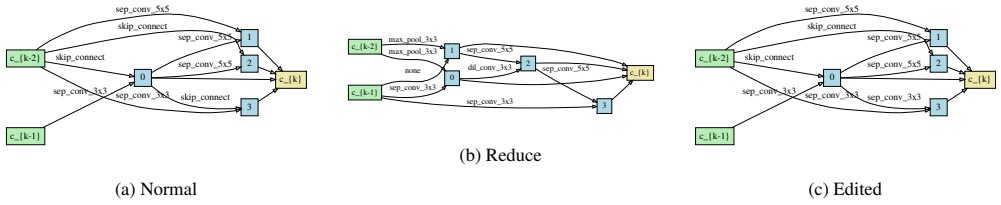

(a) Normal
(b) Reduce
(c) Edited

Figure 17: Genotypes of BANANAS architecture (White et al., 2021)

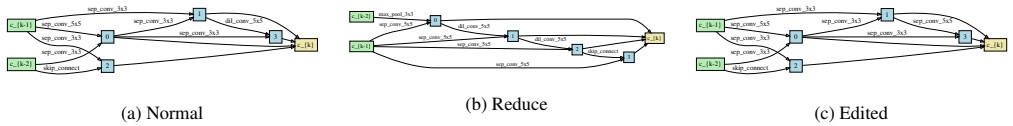

(a) Normal   (b) Reduce   (c) Edited

Figure 18: Genotypes of DRNAS architecture (Chen et al., 2021b)

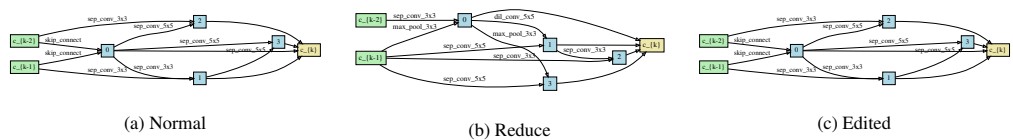

(a) Normal   (b) Reduce   (c) Edited

Figure 19: Genotypes of GAEA architecture (Li et al., 2021). Note that the edited genotype is identical to the original normal genotype as it is already compliant with both *Prim* and *Skip* constraints.

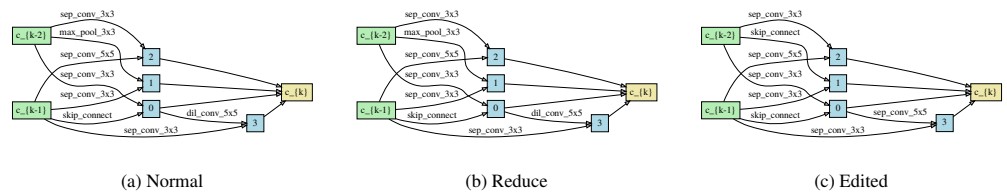

(a) Normal   (b) Reduce   (c) Edited

Figure 20: Genotypes of NASBOWL architecture (Ru et al., 2021).

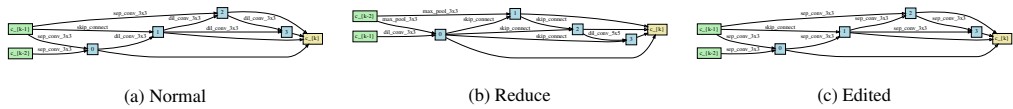

(a) Normal   (b) Reduce   (c) Edited

Figure 21: Genotypes of NOISYDARTS architecture (Chu et al., 2020)

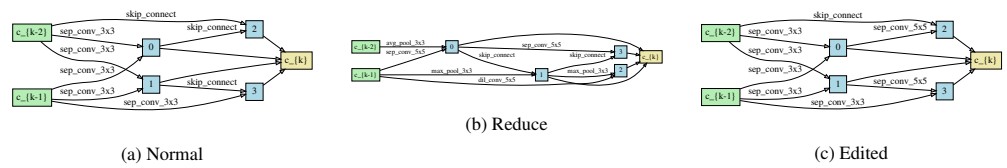

(a) Normal   (b) Reduce   (c) Edited

Figure 22: Genotypes of DARTS_PT architecture (Wang et al., 2021c)

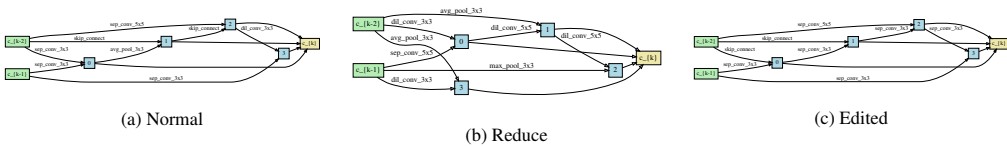

(a) Normal   (b) Reduce   (c) Edited

Figure 23: Genotypes of SDARTS_PT architecture (Wang et al., 2021c)

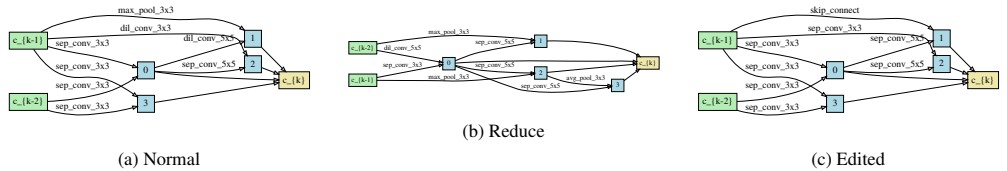

(a) Normal

(b) Reduce

(c) Edited

Figure 24: Genotypes of SGAS_PT architecture (Wang et al., 2021c)

## E.2 RANDOM GENOTYPES SAMPLED IN THE PRIMSKIP GROUP

We show some examples of the genotypes generated via the constrained random sampling in the *PrimSkip* group in Sec 4 in Fig 25, while the two architectures selected for the CIFAR-10/ImageNet experiments on the larger architectures is shown in Figs 26 and 27.

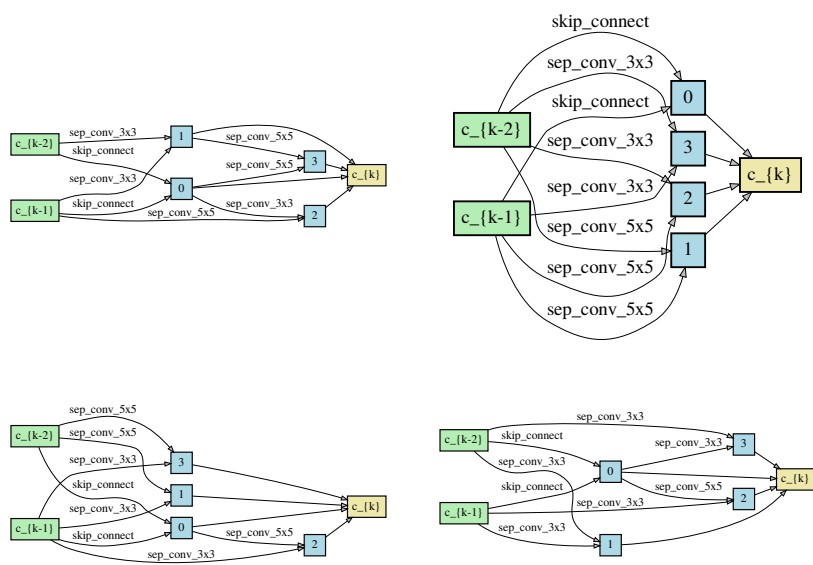

Figure 25: Some of the randomly sampled architectures in the *PrimSkip* group.

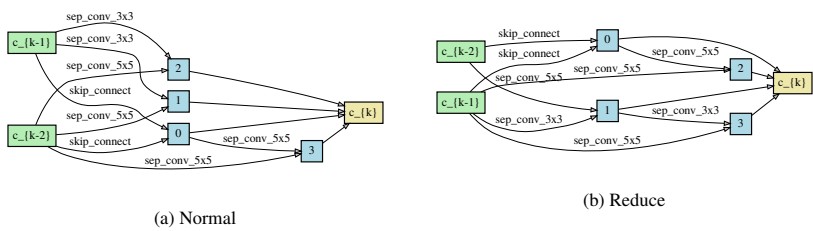

(a) Normal

(b) Reduce

Figure 26: Randomly selected *PrimSkip* architecture 1 for the experiments on the larger architectures

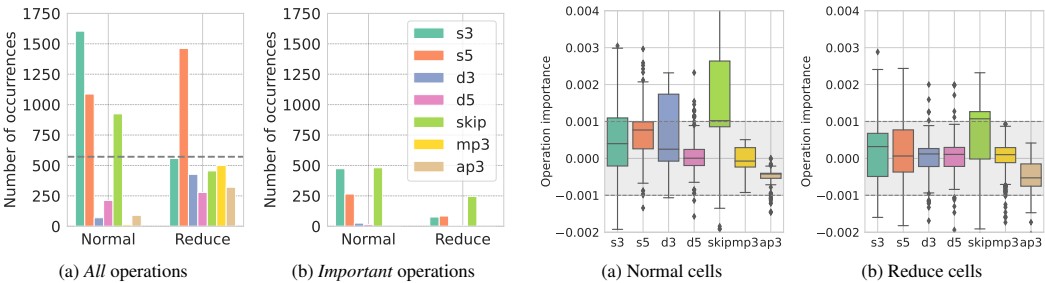

<table>
</table>

| (a) *All operations* | (b) *Important* operations | (a) Normal cells | (b) Reduce cells |
|---|---|---|---|

Figure 28: Distribution of (a) all and (b) important operations by the primitive types *according to the* BA-NANAS *surrogate*. The gray dashed line in (a) denotes the expected number of occurrences if the operations are uniformly sampled.

Figure 29: Box plots showing the distribution of the operations importance in (a) normal and (b) reduce cells *according to the* BANANAS *surrogate*. The important operations by the definition of the paper are shown outside the gray shaded area.

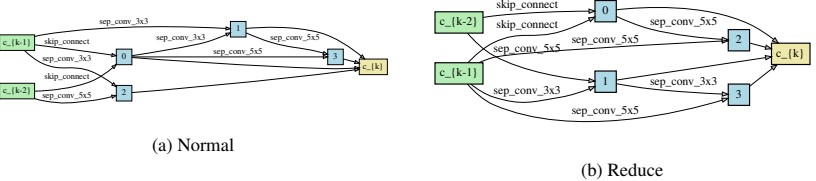

(a) Normal

(b) Reduce

Figure 27: Randomly selected *PrimSkip* architecture 2 for the experiments on the larger architectures

# F    REPRODUCING RESULTS WITH LESS TRAINING DATA

In this section, we show that it is possible to reproduce many results in the paper using less than 0.4% of the data compared to the full set of more than 50,000 architecture-performance pairs used in the NB301 surrogate, thereby motivating the use of the tools introduced in this paper as a generic, cost-effective search space inspector: For the experiments conducted, we use the surrogate from BANANAS (White et al., 2021), which combines a neural ensemble predictor with path encoding of the architectures – it is worth noting that alternative surrogates may also be used, but it is preferable to use a sample-efficient surrogate that is capable of finding meaningful relations in the input data with a modest number of evaluations. Specifically, we randomly sample 200 architectures from the search space and query their NB301 predicted performance as a proxy for the ground-truth performance. We then compute the path encoding of each architecture and train a predictor with the default hyperparameters from White et al. (2021).

We first verify whether the surrogate using less data is able to learn meaningful patterns by drawing another 200 random unseen architectures in the search space and compare the predictions by the neural ensemble predictor vs the NB301 prediction (Fig 30) and it is clear that the regression performance is already satisfactory (with a Spearman rank coefficient of 0.77) despite using much less data.

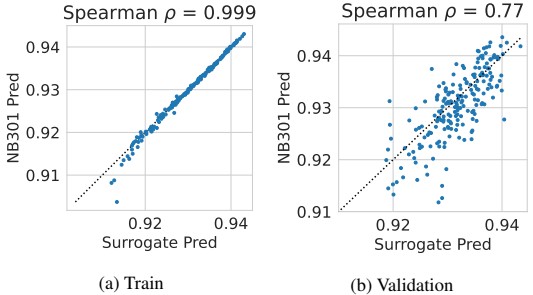

(a) Train

(b) Validation

Figure 30: Regression performance of the BANANAS predictor with 200 training data vs the NB301 surrogate with more than 50,000 training data.

We then repeat the analysis in the main text, and show the operation-level findings (Sec 3) in Figs 28 and 29 and the important subgraphs corresponding to Sec 4 in Fig 31. It is worth noting that most of

the findings are already highly similar to those in the main text, although, for example, the operation importance distributions in Fig 29 have a larger variance due to the less certain predictions. The main purpose of this study is to show that combined with the explanability tools used in the present work, an appropriate performance surrogate, which is only used as a vessel towards searching in some search methods so far, can be itself valuable. We demonstrate that it could shed insights into the strengths and weaknesses of an arbitrary search space with a modest number of observations – for example, during design of a new search space, we may randomly sample and evaluate a modest number of architectures and similarly fit a surrogate. We may then use the explainability tool to inspect the search space in a similar procedure in this section – we believe this could potentially prevent some of the pitfalls described in the existing cell-based search spaces to recur in prospective new ones during the design process.

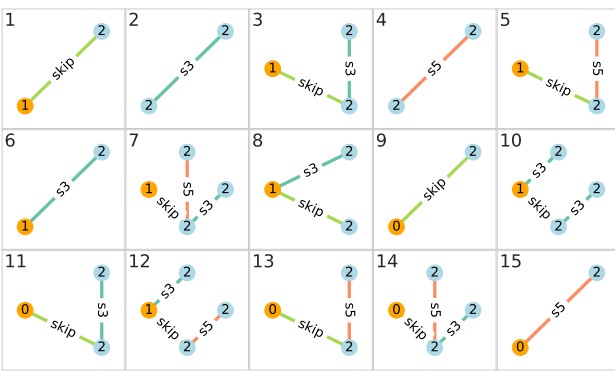

Figure 31: Frequent subgraphs in the good-performing architectures ranked by ratio of supports between the important subgraphs and the reference and properties of the discovered frequent subgraphs *according to the* BANANAS *surrogate*. Note that the residual link + separable convolution patterns are highly similar to those identified in Fig 7 in the main text

## G    PRIMITIVE REDUNDANCIES WITH MULTIPLE OPTIMISATION OBJECTIVES

Most of the findings of the paper, including the current definition of the OI, are based on maximising test accuracy as the sole objective, as the majority of the existing literature on cell-based NAS (at least those surveyed in App. C) focuses on maximising performance as the primary performance metric. However, alternative cost metrics often expressed in terms of FLOPS or number of parameters, are sometimes also incorporated as a part of the objective (e.g. maximising test accuracy while minimising the number of parameters in the resulting architecture) especially for deployment on cost-sensitive platforms like mobile devices. This give rises to a multi-objective optimisation problem, and a reasonable question to ask is that whether some of the main findings, in particular the redundancy of primitives identified in the paper, still hold, as some of the primitives redundant or unimportant for *maximising performance* might be useful for *minimising costs*. In this section, we show some preliminary results that at least in terms of cost metrics like FLOPs and/or number of parameters, the otherwise redundant primitives are still rather unimportant.

We consider the PrimSkip group of architectures described in Sec 4 since we find that they give rise to the best performing architectures in the search space. Here, to obtain the trade-off between performance and number of model parameters while still maintaining the PrimSkip constraints, instead of sampling only from {s3, s5} primitives for the operation spots (except for the fixed residual connections), we introduce another parameter $p \in [0, 1]$, which denotes the probability that the parameterless skip is chosen instead of the separable convolution primitives except the two fixed residual connections: when $p = 1$, skip will be selected for every possible operation spot, giving rise to the most lightweight possible architecture. On the other hand, when $p = 0$, we are back at the base scenario where the only skips are the 2 residual connections fixed a-priori.

To investigate the importance of the pooling operations ({mp3, ap3}), we then sample from another group of architectures that is identical to above, except that for the parameterless primitive, we may additionally choose from the pooling operations in addition to skip. For both groups, we sample 1,000 architectures from the constrained search spaces with random $p \in [0, 1]$ and show the

trade-offs in terms of 2D Pareto fronts between accuracy and number of parameters (Fig. 32)(a)) and between accuracy and FLOPs (Fig. 32(b)): if the pooling operations are indispensable to achieve good accuracy-cost trade-off, we should have observed a dominating Pareto fronts of the PrimSkip+Pool group of architectures over the PrimSkip group only. Nonetheless, we observe broadly comparable Pareto front for both groups (in terms of FLOPs, PrimSkip seems to find marginally more Pareto efficient solutions), suggesting that even after accounting for the conflicting objectives, it seems that in the existing search space design, pooling operations reduce costs largely because they are parameterless, and skip connections, which are similar parameterless, can be used to construct similarly performant/lightweight models but with reduced redundancies in primitive choices. It is nevertheless worth emphasising that this does *not* suggest that the pooling operations in general are not valuable; we hypothesise that the primary reason leading to the redundancies of pooling, as we also mentioned in the main text, is because the existing cell-based search space manually inserts pooling layers *between* cells, rendering pooling *within* cells unhelpful, for performance and/or for costs.

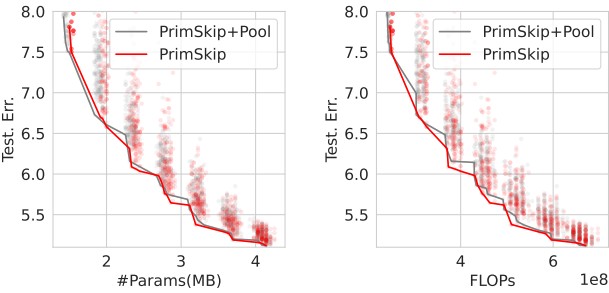

Figure 32: Test errors against number of parameters (a) and FLOPs (b) of the sample architectures in the two groups.

## H  PERFORMANCE OF NAS ALGORITHMS IN PRIMSKIP SEARCH SPACE

Another interesting experiment to consider the performance the existing NAS algorithms in the Prim-Skip space as opposed to the original DARTS search space. In this section, we conduct experiments on two search methods, each representing a dominant genre in NAS: DARTS (Liu et al., 2019) as the seminal differentiable search method, and NAS-BOWL (Ru et al., 2021), the state-of-the-art query-based NAS method. For NAS-BOWL, we run 5 repetitions with a query budget of 100 architectures. For DARTS, we use the default hyperparameters provided in the original DARTS paper to train the supernetwork for 50 epochs. We show the trajectories of test errors and the final Fig. 33 (DARTS is not query-based, so Fig 33(b) shows the test error of the architecture proposed by DARTS at the end of each epoch, as if it DARTS is terminated and discretised at that epoch. This metric is also referred to by, for example, *oracle* test error in previous works (Li et al., 2021)): it is evident that conducting search in the constrained search space leads to massive speed-ups compared to doing so in the original search space, again confirming that the simple rules identified in the paper have constrained the architectures to a very high-performing region of the search space.

## I  DETAILED EXPLANATION OF THE FREQUENT SUBGRAPH MINING PROCEDURE IN SEC 4

As mentioned in Sec 4, *support*, which measure the number of times a subgraph occurs in a set of graphs, favour simpler subgraphs as they appear more frequently "by nature": for example, consider the simplest subgraph such one consists of a single operation (e.g. s3) as a single edge. It is much more likely, even by random generation, for a graph to contain a s3 compared to another more complicated subgraph such as a triangle, with operations on 3 edges being {s3, s5, skip}. As a result, s3 will have a much larger support in the set of graphs, but it does not imply it is a more significant subgraph.

Therefore, a more interesting and meaningful metric is how much *more* over-represented (or under-represented) a subgraph is, within a set of graphs, beyond its "natural" level of occurrences. An

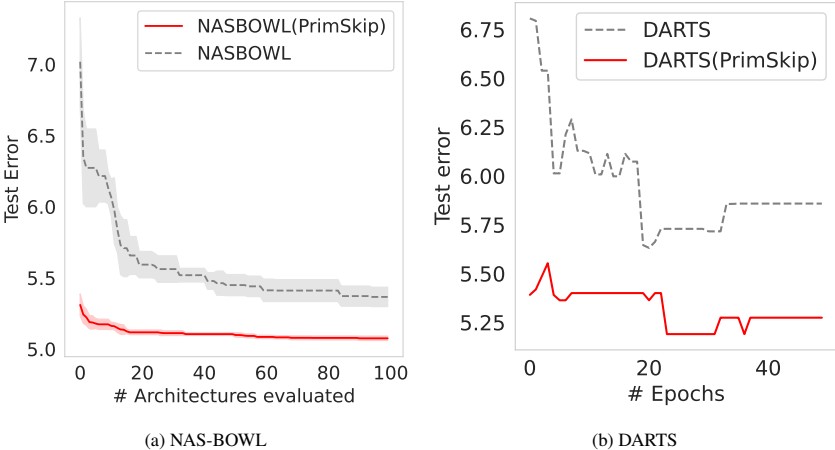

(a) NAS-BOWL

(b) DARTS

Figure 33: Performance of NAS-BOWL and DARTS in the constrained PrimSkip search space. The NAS-BOWL results shown with mean ± std over 5 random seeds.

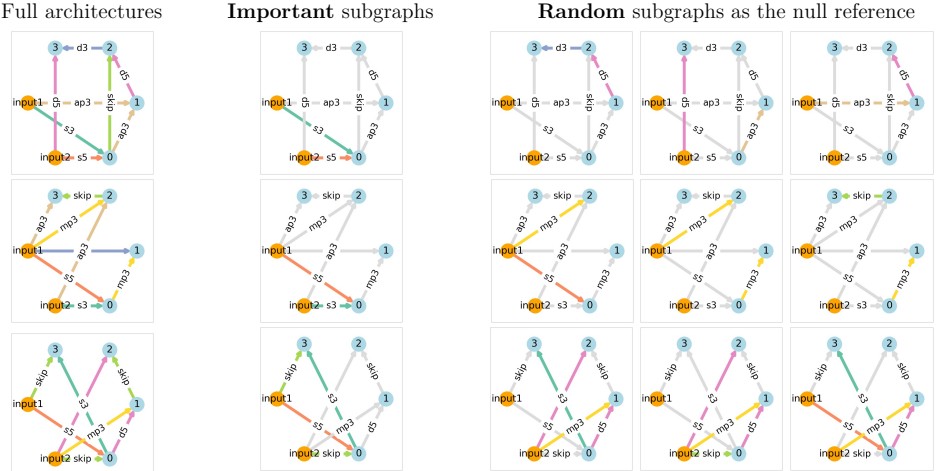

Figure 34: Demonstration of the frequent subgraph mining procedure in the important subgraphs. *Full architectures* shows the DAGs representing the entire search cells; *Important subgraphs* (the coloured subgraphs) shows the subgraphs of each full architecture consist of only the important operations (i.e. those with an OI above some threshold). *Random subgraphs* (the coloured subgraphs) act as the null reference in this case: consider a full architectures $G$ and its important subgraph $g^f$ with $m$ edges, we randomly sample exactly $m$ edges from each $G$, possibly with multiple repetitions, to form $g^r$, the corresponding random subgraph(s) of $G$.

intuitive way to measure this over-representation is thus to compute ratio between the support of the subgraphs in the *target* set and that of some null *reference* set. As shown by the demonstration in Fig 34, the target in this case is simply the important subgraphs as defined in Sec 4. The null reference, as explained in the captions of Fig 34, are constructed from the randomly sampled subgraphs of each full architecture DAG. The key intuition here is that if a subgraph has a high support simply by virtue of its simplicity or otherwise, it should have high support values in *both* the target and null reference sets and taking the ratio would cancel out its influence (due to the high denominator value). On the other hand, if there exists an underlying mechanism such that certain subgraphs are more likely to be important (which is the primary objective of the FSM in this context – to identify whether there exists sub-structures within cells that correlate with good performances), then they will likely be more over-represented in the target set than in the null reference. Indeed, the top subgraphs as identified in Fig 7 all feature rather more complicated structures and a low "natural" incidence, but they nevertheless appear as recurring patterns in the important subgraphs, suggesting that they have a strong connection to the overall good performance of the cells.

## J    SUGGESTIONS FOR FUTURE DIRECTIONS

In this section, we elaborate in detail the two promising future directions that we identify in Sec 6 that would hopefully guide future NAS researchers and practitioners to build better search spaces:

- *Simplify the cells but relax the macro-connections between cells:* Current cell-based NAS almost overwhelmingly focuses on cells but manually fix the inter-cell connections in a linear manner similar to the classical networks (e.g. ResNet) that are known to do well. On the other hand, other search spaces like the MobileNet space, are arguably more flexible but are purely macro-based – in our opinion, there could be potential to preserve the convenience of the search cell, but at the same time to give more freedom on how different cells are connected. By relaxing the constraints on how the different cells may be connected, there is not only a potential for a much larger and expressive search space, but also be a greater variability in overall graph theoretic topological properties (as opposed to topological properties *within cells only*) and widths/depths of the resulting networks, which have been shown to significantly influence the performance (Xie et al., 2019a; Ru et al., 2020). We leave the detailed specification of such a search space to a future work, but a concrete direction would be to further investigate and improve on the hierarchical search spaces such as the one proposed in Ru et al. (2020).

- *Towards lower-level searching:* As hypothesised in the main text, the current search methods might be implicitly encouraged to find something similar to the existing architectures, as the current NAS pipelines often borrow heavily from the classical network designs (e.g. the linear macro connections between cells and the manually inserted conv and pooling layers are often found in the manually-designed networks). While this is not necessarily bad on its own, we might be making it more difficult for search methods to find novel architectures beyond what is already known. An alternative could be that instead of searching on high-level primitives and operators, we could search on something more fundamental like searching for sequences of mathematical operations. The pioneering work is Real et al. (2020), but similar to some early-stage NAS works, it was built on evolutionary algorithms, which are not particularly sample- and computation cost-efficient (as opposed to later works like one-shot/differentiable NAS which dramatically reduced computing cost of NAS). Thus, another promising direction would be to explore searches on a more fundamental level that is free of bias from existing design, but to develop more scalable and tailored solutions to make the problem more tractable.

- *NAS Benchmarks beyond simple cells:* The advent of the NAS benchmarks have greatly democratised NAS and has partially led to the booming of recent NAS research. While exceptions may exist, the most popular NAS benchmarks on computer vision problems, which greatly facilitate faster iteration of search methods, are currently based on the cell-based spaces (NAS-Bench-101/202/301/x11) (Ying et al., 2019a; Dong & Yang, 2020; Siems et al., 2020; Yan et al., 2021). An improved search space would be made more widely accepted in a much easier way if a suitable benchmarking tool is developed. Therefore, we argue there is a need is to build NAS benchmarks beyond the current cell-based spaces.

