# OpenReview forum: "On Redundancy and Diversity in Cell-based Neural Architecture Search"
_ICLR.cc/2022/Conference — ICLR 2022 Poster_

### Official Review · Reviewer_2nKR · 2021-11-02

**Correctness:** 3
**Technical Novelty And Significance:** 2
**Empirical Novelty And Significance:** 2
**Recommendation:** 5
**Confidence:** 4

**Main Review:**

Strengths:
1. Reducing the complexity of search spaces in NAS is a strong motivation.
2. The authors identify multiple interesting redundancies that may help influence future designs.
3. Paper is reasonably easy to read, although numerous grammatical errors (see comments for just the first part).

Weaknesses:
1. It is not at all clear how a result like this generalizes beyond CIFAR. While it is clear that, if the goal is to search for the best architecture on CIFAR, then reducing the search space in this way is useful. However, the point of CIFAR is really just to be a testing ground for algorithms run on other tasks. On those it is entirely unclear that the less useful operations will still not be useful; for example, larger dilations are good for sequence modeling while max-pooling is important for local invariance. The focus on CIFAR is interesting as a baseline but does not give evidence that these restrictions would work on other datasets.
2. Another missing component is the question of compute time/energy use, which depends on the topology and operations. Many search spaces focus on more efficient architectures, which may themselves be more redundant in the full space and thus will be ignored in this study.

Comments:
1. [*This simplification reduces the search space (but still highly complex),*] Unclear parenthetical.
2. [*it is fair to state that such cell-based spaces currently dominates.] -> dominate
3. [*high-level patterns only (Shu et al., 2020; Zela et al., 2020) We*] Missing period.
4. [*NAS paradigms such as transformers*] -> Transformers
5. [*Note that we may not obtain NB301 performance prediction on these architectures, as NB301 requires all 16 operations to be enabled with valid primitives.*] This does not make sense to me. If NB301 contains all architectures in DARTS, surely it contains those that do not have any of the removed operations.
6. [*Fig. 4 shows that the OI distribution across all primitives are centered close to zero in reduce cells: both suggest that reduce cell is less important to the architecture performance.*] Unclear how this shows the reduce cell is unimportant, and not that it contains redundant operations.

**Summary Of The Paper:**

Summary:
This paper studies the DARTS search space using predicted results from NAS-Bench-301 and presents several findings that suggest that SOTA performance on CIFAR-10 can be obtained via a relatively small subset of the space. They leverage these findings to suggest a simplified search space in which random architectures achieve strong performance and also to give suggestions for future development of NAS algorithms.

**Summary Of The Review:**

While the discoveries in the paper are interesting and may be useful for searching CIFAR, it is entirely unclear how they extend to more practical NAS settings, e.g. tasks beyond CIFAR and constraints beyond just the DARTS search space. Indeed the findings may not be relevant if they do not hold in other settings NAS may be applied to. I view answering these questions as critical and so lean against acceptance.

---

> ### Author Response · Authors · 2021-11-15
> **Thank you for your feedback (1/2)**
>
> We thank the reviewer for their constructive feedback! Please see our overall response and the detailed reply below which we hope would resolve any concerns that the reviewer might have; we hope that in light of this the Reviewer would consider increasing their score. We'd also like to thank the reviewer for identifying the typo/grammar errors in particular; these have been corrected in the rebuttal revision, and the reply below mainly deals with the major technical comments the reviewer had:
>
> > 1. Unclear how results generalise beyond CIFAR
>
> While we reach our key findings using the DARTS space in CIFAR-10 since that is the only option provided by the NAS-Bench-301 (otherwise most of the verification would require a huge amount of computational resources beyond our capability), we do test and verify that our key results generalise beyond the CIFAR-10/DARTS space combination:
>
> - **Table 2b shows results on the full ImageNet**, the most difficult and largest-scale amongst the datasets commonly considered by NAS papers: although Prim and Skip constraints are derived from our analyses on CIFAR10, they are transferable/generalisable to ImageNet, as we can see that the 2 random PrimSkip architectures perform on par, if not better, than the SoTA searched directly on ImageNet. (note that the Prim and Skip constraints, which force architectures to 1) choose from only 3 primitives, 2) have normal and reduction cells sampled using the identical rule, and 3) to contain the residual link, encode most of the findings on search space redundancy and diversity in the paper).
>
>     If *any* of the major findings we had on CIFAR-10 does not extend to ImageNet (e.g. if extra primitives or searching reduce cells separately is actually important in ImageNet; or if there exists some pattern beyond the simple sep conv + residual link that is critical for performance on ImageNet but not on simpler datasets), then it is unlikely to see sampled PrimSkip architectures perform so competitively compared to the SoTA. Thus, even if we do not *derive* these findings on ImageNet directly, we argue the tests we conduct suffice in showing that the patterns hold true also on ImageNet.
>
> - App D shows analysis on NB201 (a related but different cell-based space to NB301/DARTS) on **CIFAR-10, CIFAR-100 and the downsampled ImageNet**. The main findings also hold here.
>
> Therefore, we argue that the key findings of the paper (that we have redundant primitives and performance hinges on a few, non-diverse key features) do extend beyond CIFAR to at least the other common datasets considered in the NAS literature.
>
>
> > 2. [Note that we may not obtain NB301 performance prediction on these architectures, as NB301 requires all 16 operations to be enabled with valid primitives.] This does not make sense to me. If NB301 contains all architectures in DARTS, surely it contains those that do not have any of the removed operations.
>
> This is a design decision taken by the NB301 authors, not us: It is possible for an architecture to contain some disabled operations, but it was empirically observed that good-performing architecture almost never contains a disabled operation (i.e. it is possible to improve performance further by replacing the disabled operation(s) with some enabled operation(s) -- however, the experiment in question of our paper is to determine the functional part of the cell, not to find the absolute best architecture), so the NB301 authors made the decision to force the architectures to contain no disabled operations. In this sense, considering these architectures that are known to be sub-optimal, the NB301 space is smaller than the full DARTS space, although for practical purposes they are the same. This claim is also made in the original NB301 paper: *“We now describe the NAS-Bench-301 dataset which consists of ∼60k architectures and their performances on CIFAR-10 (Krizhevsky, 2009) sampled from the most popular NAS cell search space: the one from DARTS“*.
>
> *-- to be continued below --*

---

> > ### Author Response · Authors · 2021-11-15
> > **Thank you for your feedback (2/2)**
> >
> > *-- continued from above --*
> >
> > >3. [Fig. 4 shows that the OI distribution across all primitives are centered close to zero in reduce cells: both suggest that reduce cell is less important to the architecture performance.] Unclear how this shows the reduce cell is unimportant, and not that it contains redundant operations.
> >
> > If all primitives (even including the ones that are deemed to be important in the normal cells) of the reduce cell have OI close to zero (meaning that they may be shuffled with any other primitives with no or little performance deterioration), the conclusion is that the reduce cell as a whole is less important. We also **explicitly** verify this is the case in Fig 6 where we actually trained the architectures (i.e. not the NB301 predictions): setting reduction cells to be identical to the normal cells (red <- nor) led to no statistically significant deterioration in performance, and even setting the entire reduce cell to be skip connections (red<-skip) only led to minor deteriorations. If the reduction cells are critical to the performance, such shuffling should  impact the performance a lot, at least to a comparable extent when we shuffle the normal cells, but this is not the case. As such, the conclusion is that even though reduction and normal cells are treated equally during search (in terms of dimensions to be searched), their contribution to the final performance are not equal; under the current cell-based search space, we can reach similar performance by searching for one cell structure only and applying it on both normal and reduction -- this is shown both in Fig 6 and verified experimentally in Table 2: The normal and reduce cells of PrimSkip are sampled using identical rules (i.e. not seached separately), yet they perform similarly to the SoTA (which searches both) in both CIFAR-10 and ImageNet.
> >
> > > 4.Another missing component is the question of compute time/energy use, which depends on the topology and operations. Many search spaces focus on more efficient architectures, which may themselves be more redundant in the full space and thus will be ignored in this study.
> >
> > The reviewer is referred to our Point 4 of our response to Reviewer 7BWB (https://openreview.net/forum?id=rFJWoYoxrDB&noteId=uDV3-tmJHwV). In short, even when accommodating the possible dual objective of maximising performance and minimising cost (in terms of #params and FLOPs, two very common metrics measuring general computational burden), under the current design we still have many redundancies.The reviewer is referred to App. G in the updated paper for detailed discussions.

---

> > ### Comment · Reviewer_2nKR · 2021-11-22
> > **Response**
> >
> > Thank you for the response. My concern is not that the search space restrictions weren't as verified on ImageNet as they were using NB301 on CIFAR, but that this task-specific approach to search space reduction flies in the face of the goal of NAS, which is to automate architecture design. If we consider a scientist faced with a new task, NAS suggests that they first construct a search space and then run a search algorithm to find a good architecture. Unless the scientist has an unrealistic level of knowledge about the types of architectures that work well on the task (such as the kind provided by NB301 for CIFAR) they will almost certainly have lots of redundancy in their search space; the goal of NAS researchers is then to design algorithms that can effectively search any such space (within reason) and find a good architecture. Indeed, your Table 2a results show that the most recent algorithms (especially GAEA but also BANANAS and DrNAS) are already effective at doing so, or close to it, at least when searching the DARTS space on CIFAR-10.
> >
> > However, your conclusion is not "NAS algorithms are great, they can navigate complex redundant search spaces and find good architectures" but that your "findings cast doubts into our ability to discover truly novel architectures in the existing cell-based search spaces." In fact I agree with you on the last point but I don't believe your paper has actually provided evidence for it. What your paper does show is that by carefully analyzing the search space our scientist might make it easier for the search algorithm to find a good architecture. However, careful analysis is *not* automation and thus is counter to the goal of NAS. Perhaps if the results of the analysis did not vary from task to task this approach might be convincing, but this paper has only (possibly) shown this to be true on a few very closely related image classification tasks. The result would at least have stronger evidence if it held on a large number of diverse tasks or search spaces, such as those from TransNAS-Bench-101 or NAS-Bench-360.

---

> > > ### Author Response · Authors · 2021-11-23
> > > **Response to your reply (1/2)**
> > >
> > > Thank you for your reply.
> > >
> > > > this task-specific approach to search space reduction flies in the face of the goal of NAS, which is to automate architecture design. … However, careful analysis is not automation and thus is counter to the goal of NAS.
> > >
> > > We do not believe that the purpose of this paper is counter to the goal of NAS: similar to the motivations of multiple previous works that aim to enhance the understanding of search spaces via thorough analysis (cited in Related Works), we too argue that understanding the search space and the architectures discovered is in no way antithetical to NAS, but has a strong potential to benefit it and to inform us on what features the algorithms have discovered (which could potentially inspire researchers even beyond the NAS community).
> > >
> > > **We would also like to strongly reiterate that this paper is not about proposing “search space reduction” on known search spaces to accelerate search methods**, as the reviewer claimed to be a key message of the paper; the reduction is just a by-product of our analysis on the existing search spaces only, and it is a phenomenon we in fact consider as sub-optimal since this implies that the architectures discovered are non-diverse. On the very contrary, in Sec 6 and App J, our proposed goal is to provide suggestions on how we might improve future search spaces so as to alleviate some of the issues we identify in the studied space, and make the search space even more expressive and diverse, not “reduced” (such as how we could simplify cells but to relax macro connections among cells and do lower-level search).
> > >
> > > > However, your conclusion is not "NAS algorithms are great, they can navigate complex redundant search spaces and find good architectures" but that your "findings cast doubts into our ability to discover truly novel architectures in the existing cell-based search spaces." In fact I agree with you on the last point but I don't believe your paper has actually provided evidence for it. Perhaps if the results of the analysis did not vary from task to task this approach might be convincing, but this paper has only (possibly) shown this to be true on a few very closely related image classification tasks.
> > >
> > > While it is possible to interpret the present analysis as “task-specific”, it is worth noting that the overwhelming majority of the current NAS methods, including the GAEA, BANANAS and DrNAS referred to by the reviewer, are all “task-specific” as people often execute NAS for a very specifically defined task. In fact, all papers in App. C consider image classification, and very often, on image classification only -- including GAEA, BANANAS, DrNAS, and **21 out of the 24 recent papers published in top machine learning venues surveyed in App C** (the 3 that indeed consider additional tasks (SemiNAS, Few-shot NAS and CafeNet), nevertheless still consider the image classification task). In fact, even in the Trans-NAS-Bench-101 paper raised by the reviewer, the authors remarked in Sec 5: *“Because the field of transferable NAS is new and nascent, there are only limited number of transferable architecture search algorithms''* -- but clearly just because most current NAS methods are “task-specific” and use image classification as the prime example, it does not mean that there is no evidence for their effectiveness.  Consequently, given that we aim to do a post-hoc analysis upon the **architectures discovered by existing methods and search spaces**, we feel that our analysis focusing on image classification is justified and our coverage of CIFAR & ImageNet is on par with the current norm in NAS. Within the image classification task, we do argue that we have provided ample evidence supporting the claims in the paper, a point that is also agreed upon by all other reviewers.
> > >
> > > In summary, while we do agree that generalizing our findings beyond the current domain can be an important and interesting future direction, we argue it is not fair to claim that “we have not provided evidence” simply because we do not consider domains outside image classification, given that most cell-based NAS methods themselves provide evidence on image classification only, which is still a task with a huge amount of resources and research attention committed and there is still plenty to learn *in this dominant, most studied task in NAS* alone.
> > >
> > > *-- to be continued below --*

---

> > > > ### Author Response · Authors · 2021-11-23
> > > > **Response to your reply (2/2)**
> > > >
> > > > *-- continuned from above --*
> > > >
> > > > > If we consider a scientist faced with a new task, NAS suggests that they first construct a search space and then run a search algorithm to find a good architecture. Unless the scientist has an unrealistic level of knowledge about the types of architectures that work well on the task (such as the kind provided by NB301 for CIFAR) they will almost certainly have lots of redundancy in their search space; the goal of NAS researchers is then to design algorithms that can effectively search any such space (within reason) and find a good architecture
> > > >
> > > > We do not deny that any new search space is likely to be redundant and the search method should aim to work well nevertheless. However, **acknowledging the existence of redundancies and/or other drawbacks does not mean we cannot or should not seek to understand them and to act upon them**.  On the contrary, we believe that just like how NAS algorithms iterate (e.g. from the earlier RL-based NAS to the modern, one-shot NAS), the NAS search spaces should also evolve to enable continual progress of NAS to come up with novel, high-performing architectures for various scenarios, and for this matter, understanding the pros and cons of the current search space is key, and this is also what we believe where a gap exists in the literature.
> > > >
> > > > Our paper aims to shed insight on this by not only analysing the present search space, but also by presenting an analysis framework to understand any search spaces (App F): we show that at least for image classification tasks, surrogate models learnt by NAS can be used in our framework to cheaply obtain insights on the importance of operations and key patterns of the search space, so NAS practitioners can use our proposed tools to analyse their search spaces, even without “unrealistic level of knowledge”.
> > > >
> > > > Likewise, **just because existing NAS methods work reasonably well in the current spaces does not mean that these search spaces are optimal as benchmarks and that they cannot be improved upon**: what we’ve seen is that the existing cell-based methods have developed on a highly-redundant yet non-diverse search space only, each yielding highly similar architectures with very close performance, which we believe is at least sub-optimal for the future, continual development of NAS as they 1) provide little insight on the relative performance of the methods beyond the benchmark and 2) might not enable search methods to find diverse, novel solutions. We argue it is far better to have improved search spaces as benchmarks that enable fair comparison *while* also meeting other desiderata.

---

> > > > ### Comment · Reviewer_2nKR · 2021-11-27
> > > > **Response**
> > > >
> > > > Thank you for the detailed response.
> > > >
> > > > 1. [*similar to the motivations of multiple previous works that aim to enhance the understanding of search spaces via thorough analysis (cited in Related Works), we too argue that understanding the search space and the architectures discovered is in no way antithetical to NAS, but has a strong potential to benefit it and to inform us on what features the algorithms have discovered*]
> > > > - I have no issue with papers trying to understand search spaces and architectures by NAS algorithms. Indeed, the cited papers do largely identify issues (notably not always search space issues but also search algorithm issues) with applying NAS to image classification and propose fixes that are evaluated on image classification tasks; I think this is fine, although certainly including evaluations beyond image classification would be an improvement. However, the current paper identifies things that are *not* issues on image classification—-it largely seems NAS algorithms successfully navigate the observed phenomena—and then proposes fixes that are not evaluated at all. An excellent contrast is the cited paper by Zela et al. (2020), which uses CIFAR-10 to identify an issue (DARTS returning all skip-connections) that would very believably also be problematic on other tasks; they then propose a fix that is evaluated not just on image classification but also on disparity estimation and language modeling. On the other hand, the issue identified in the current paper does not seem to be a major problem on CIFAR-10 and it is not as evident that it would be a major problem on other tasks.
> > > >
> > > > 2. [*this paper is not about proposing “search space reduction” on known search spaces to accelerate search methods, as the reviewer claimed to be a key message of the paper; […] our proposed goal is to provide suggestions on how we might improve future search spaces so as to alleviate some of the issues we identify in the studied space, and make the search space even more expressive and diverse, not “reduced”*]
> > > > - Fair, I was incorrect about the motivation of the work in my original review. Notably, (a) this means your quoting of my “strong motivation” strength as a positive is a bit problematic as it is founded upon my misreading and (b) the fact that the search algorithm “randomly sample from PrimSkip” works well is the best-verified practical consequence of the paper.
> > > >
> > > > 3. [*While it is possible to interpret the present analysis as “task-specific”, it is worth noting that the overwhelming majority of the current NAS methods, including the GAEA, BANANAS and DrNAS referred to by the reviewer, are all “task-specific” as people often execute NAS for a very specifically defined task.*]
> > > > - Covered by my response to point 1 above.
> > > >
> > > > 4. [*acknowledging the existence of redundancies and/or other drawbacks does not mean we cannot or should not seek to understand them and to act upon them. On the contrary, we believe that just like how NAS algorithms iterate […] the NAS search spaces should also evolve to enable continual progress of NAS to come up with novel, high-performing architectures for various scenarios*]
> > > > - I agree but do not believe the paper has made a significant contribution in this direction. As a comparison I can point to two existing papers I am familiar with from the NAS community that work on diversifying search spaces. The first which you cite by Real et al. (2020) comes up with an incredibly general setup where even the training procedure is discovered; being able to fruitfully search this space at all was the significant contribution. The second by Roberts et al. (*Rethinking neural operations for diverse tasks*, NeurIPS 2021) demonstrates specific failures of current search spaces such as DARTS and provides a new search space that seems to avoid them. As far as I can see the current paper neither demonstrates significant failures (in terms of error/latency metrics) nor proposes an approach that is then empirically verified, except to use surrogate models for future search spaces. The latter I discuss in point 5 below.
> > > >
> > > > 5. [*we show that at least for image classification tasks, surrogate models learnt by NAS can be used in our framework to cheaply obtain insights on the importance of operations and key patterns of the search space, so NAS practitioners can use our proposed tools to analyse their search spaces, even without “unrealistic level of knowledge”*]
> > > > - The paper claims that 200 architectures are required to reproduce most of the results found by analyzing NB301; I think it is at least questionable that this is a realistic level of knowledge given that it takes >5K GPU-hours on a V100. Training a performance predictor is basically a NAS algorithm, although perhaps a low-fidelity one could be used or the findings could be transferred to related problems. In any case this finding, which seems to be the main empirical evaluation of a proposed approach in the current paper, is relegated to the appendix.

---

> > > > > ### Author Response · Authors · 2021-11-29
> > > > > **Final Response Part III: Response to other comments**
> > > > >
> > > > > > Indeed, the cited papers do largely identify issues (notably not always search space issues but also search algorithm issues) with applying NAS to image classification and propose fixes that are evaluated on image classification tasks; I think this is fine
> > > > >
> > > > > We argue that the findings discovered in the present paper (redundancy and non-diversity) should also extend to other tasks with similar cell-based designs, given that it is a search space-level issue and as discussed, just because an issue does not affect the accuracy of image classification (or indeed performance metric of any task) does not mean it isn’t important.  Even though there is some uncertainty due to extrapolation into other tasks, we believe this is true for the vast majority of existing works that use image classification as the prime task (search methods and analysis papers included), and we argue we should not be penalised in particular on this point.
> > > > >
> > > > > On a related note, while we thank the reviewer for bringing up multiple related papers, and we will certainly discuss them when we next revise our paper, the reviewer is reminded that NAS-Bench-360, Trans-NAS-Bench-101 and the NeurIPS 2021 paper are all concurrent to our paper by the ICLR policy: NAS-Bench-360 appeared in Arxiv after ICLR deadline and Trans-NAS-Bench-101, published in CVPR (19-25 Jun), is after the cutoff date by the ICLR reviewer guidelines (*if a paper was published (i.e., at a peer-reviewed venue) on or after June 5, 2021, authors are not required to compare their own work to that paper*), and therefore we should not be penalised for not conducting experiments on or not comparing against these papers.
> > > > >
> > > > > >The paper claims that 200 architectures are required to reproduce most of the results found by analyzing NB301; I think it is at least questionable that this is a realistic level of knowledge given that it takes >5K GPU-hours on a V100
> > > > >
> > > > > Firstly, 5K GPU-hours is a massive exaggeration: on Page 2 of [3], *since each architecture evaluation takes less than a second to query from the surrogate, compared to 1-2 hours for training an architecture*, so training 200 architectures takes at most 200-400 GPU hours, more than an order of magnitude lower than 5,000. If we shorten the training to, say, 50 or even fewer epochs (which are shown to be fine in some previous works [4][5]), the cost can be further reduced to ~100 GPU hours.
> > > > >
> > > > > Secondly, we believe the said computational cost, while in no way trivial, is not exorbitant especially given the benefits and that if someone is designing a search space/developing certain search methods, the cost they have to incur anyway could be already much higher: e.g. making each of NB101/201/301 requires thousands if not tens of thousands of GPU hours, and running query-based NAS like NAS-BOWL/BANANAS often involve training of 150-200 architectures anyway (we can then just reuse the performance surrogate for analysis). In this way, there could be enough trained architectures available that we may use already.
> > > > >
> > > > > Lastly, training a performance predictor is not just “a NAS algorithm”, as the reviewer suggested. Up to this point, apart from very limited previous work (perhaps only [4] but they are constrained to 1-WL features -- see discussion in Related Works), predictor is only used for search and not used for explanation. Using a predictor as a proxy to understanding search space in the way described is novel to this work.
> > > > >
> > > > > ## References:
> > > > > [1] NAS Evaluation is Frustratingly Hard. ICLR 2020.
> > > > >
> > > > > [2] Understanding Architectures Learnt by Cell-based Neural Architecture Search. ICLR 2020
> > > > >
> > > > > [3] NAS-Bench-301 and the Case for Surrogate Benchmarks for Neural Architecture Search
> > > > >
> > > > > [4] Interpretable Neural Architecture Search via Bayesian Optimisation with Weisfeiler-Lehman Kernels. ICLR 2021.
> > > > >
> > > > > [5] NAS-Bench-201: Extending the Scope of Reproducible Neural Architecture Search. ICLR 2020.

---

> > > > > ### Author Response · Authors · 2021-11-29
> > > > > **Final Response Part II: An issue in NAS does not need to be a “significant failure” in performance metrics for it to be interesting, relevant and practically impactful.**
> > > > >
> > > > > ## On what constitutes an issue in search space
> > > > >
> > > > > The reviewer seems to suggest that an issue only matters if it concerns the error/latency metric of the resultant architecture. Unfortunately, we have to disagree on this: related to the previous point and supported by [1][2], we argue that **just because search methods “successfully navigate the observed phenomena” does NOT necessarily mean that the search space is free from problems, nor should merits of search space be judged purely on the presence or absence of performance metrics (accuracy/latency) issues alone**.
> > > > >
> > > > > In our opinion, unlike a single architecture where performance metrics are the only things that matter, the desiderata of a well-designed search space are much more multifaceted, and discussion related to *any* of these points is interesting, as it has potential practical implication on NAS search space design:
> > > > >
> > > > > 1. How diverse it is and is there a possibility for us to find novel solutions in it?
> > > > > 2. Does searching on it give us meaningful information on how search methods compare in general (e.g., an alternative, potentially more complicated search space)?
> > > > > 3. Performance metrics (e.g. accuracy/latency)
> > > > >
> > > > > Given this, we argue that a search space can have no performance issue at all but still deeply problematic. Here we give an extreme but still explanatory example: consider a singleton “search space” consisting of ResNet50 only. Any search method will instantly “find” the “optimal” architecture and thus “navigate successfully”. Since the model found is known to perform well, it will have an excellent accuracy, latency, or any other performance metric. However, we can all agree that this is far from optimal, as a search space. While “searching” on such a space will not subject us to problems like the collapse into skip connections, or indeed any performance problems, it does not provide any valuable insight from a NAS perspective, fails both Desiderata #1 and #2 and is thus far from optimal as a NAS search space.
> > > > >
> > > > > This is an extreme example of “non-diversity” but illustrates our point. Consequently, **we argue that although our paper does not identify performance issues, it offers a novel interpretable analysis framework for the search space, which helps explain the findings such as the lack of diversity in current DARTS space, as well as the similarity between the manual designed networks and the architectures discovered by highly sophisticated NAS methods. This forms one of the key contributions of our paper, and is definitely not less important and relevant**. As shown in our paper, the current spaces mainly falls short in Desiderata #1 and partly in #2.  In fact, we believe such issues, being less obvious than problems such as performance collapse, can be even more problematic. Given that DARTS is often taken as a proxy for a real-life, open-domain search space (i.e. not just a close-domain benchmark) to iterate NAS methods, many papers still use final DARTS + CIFAR-10/ImageNet test accuracy as a primary (or only) performance gauge, although as we’ve shown, the architectures discovered are more or less similar; the minute performance gap between the architectures discovered by various methods might not properly reflect the relative difference of the search methods. More fundamentally, we argue that it does not make sense to devote a vast amount of computing resources just to rediscover what we’ve already known *without being aware of this fact*, and this is due to the specific design of the search space. Thus, as we mentioned in the suggestions, e.g., we can save the search budget from optimising the cells and spend them on searching for macro-connection among the cells or micro-operations that form the primitives in the cell. We believe these can all lead to “practical consequences” in future NAS works.

---

> > > > > ### Author Response · Authors · 2021-11-29
> > > > > **Final Response Part I: Outlining our disagreements, and why our contributions are worthy**
> > > > >
> > > > > We deeply appreciate the reviewer for engaging in an active discussion with us. As the discussion period is coming to a close, please bear with us for the long final response, which we believe will be beneficial for both our discussions and for all future readers:
> > > > >
> > > > > > the fact that the search algorithm “randomly sample from PrimSkip” works well is the best-verified practical consequence of the paper
> > > > >
> > > > > > As far as I can see the current paper neither demonstrates significant failures (in terms of error/latency metrics) nor proposes an approach that is then empirically verified, except to use surrogate models for future search spaces
> > > > >
> > > > > > On the other hand, the issue identified in the current paper does not seem to be a major problem on CIFAR-10 and it is not as evident that it would be a major problem on other tasks.
> > > > >
> > > > > > However, the current paper identifies things that are not issues on image classification it largely seems NAS algorithms successfully navigate the observed phenomena
> > > > >
> > > > > We believe that the questions of “what constitutes an issue of a search space” and “what makes a worthy contribution” form the bulk of our disagreement, and please allow us to make our points clear below:
> > > > >
> > > > > ## On contributions:
> > > > >
> > > > > We do not agree that it is fair to reduce our contributions to “use surrogate models for future search spaces” and “randomly sample from Primskip works well”. The former is one but not the only contribution (elaborated later), while the latter, as we have already clarified, is a strong evidence demonstrating the effectiveness of our analysis framework on identifying important primitives while double-confirming the lack of diversity in current search space. It is, however, NOT a key message or an endorsement from us. The key message is to make the community aware of the shortcomings of the existing search spaces (note that shortcomings do not necessarily manifest themselves in problems in accuracy/latency/other performance metrics only -- see Part 2 of our reply), and to act upon them for continual development of NAS, and on this matter "PrimSkip samples perform well" is certainly not the only practical implication that could be drawn from our paper (more to be elaborated in Part 2).
> > > > >
> > > > > While we agree that AutoML-Zero and Zela et al, 2020 papers are very valuable, we disagree that there is a singular way to make contributions or that a paper whose main contribution is to make the community aware of problems on popular ideas and settings and to call for improvements is unworthy. We further argue that we should not be overly penalised for not having “empirically verified” suggestions, given that a full investigation of each suggestion probably merits a full paper on its own. In any case, we emphasise that all key findings (not just “samples from PrimSkip perform well”) are rigorously verified empirically, and final suggestions, although yet verified extensively, are linked with and inspired/motivated by  our findings. This is also agreed upon by all other reviewers including Reviewer 7BWB who was originally negative on this point. In fact, there are many papers that fit into this category that the reviewer might label as *“neither identify significant failures in terms of error/latency metrics nor propose an approach that is empirically verified”* yet proven to be impactful/inspirational -- as we argue in Part 2, we believe an issue does not need to be “significant failure in metrics” for it to be interesting and relevant. Here we give examples:
> > > > >
> > > > > [1] discover various problems related to NAS architecture ranking and evaluation. The paper mainly exposes issues, and none of the issues discovered is strictly “significant failures” in terms of performance metrics -- search methods find architectures with good accuracy despite rank disorder, and methods with additional augmentation even perform way stronger. However, this does not preclude the paper from being one that is impactful in making us aware of these problems and subsequently motivating the community to come up with fixes.
> > > > >
> > > > > [2] find that cell-based NAS methods favour shallow and wide cells and thus might miss out deep and narrow cells which can generalise better after longer training. Again, there is neither “significant failure” nor a “major problem” in terms of performance metrics as the shallow and wide cells still perform satisfactorily, but this does not mean the finding is irrelevant for future NAS developments.

---

> > > > > > ### Comment · Reviewer_2nKR · 2021-11-30
> > > > > > **Response**
> > > > > >
> > > > > > Response to most of the major points made:
> > > > > > 1. I did not reduce the contribution to "use surrogate models for future search spaces” and “randomly sample from PrimSkip works well,” I merely noted that these two are the only concrete suggestions with empirical backing. In general I don't disagree that the key findings are rigorously verified empirically, just that there are not many experimentally tested specific action items that are part of the main contribution.
> > > > > > 2. From my recollection and reading of the abstract, the main contribution of [1] was in fact a major problem with performance metrics - baselines also performed well on them and they were being unduly influenced by the evaluation protocol. My subjective view is that the issues identified in [2] would on their own not merit acceptance, but the theoretical analysis was interesting enough to warrant it.
> > > > > > 3. I don’t disagree that the DARTS space is non-diverse in the sense you’re describing, but to me this seems somewhat banal, in the sense that I don’t think it needs a full-length paper to state. Indeed, the AutoML-Zero paper largely takes it as given. Therefore my focus is largely on what practical consequence there are of the results shown here. Furthermore, at the end of the day the non-diversity of a search space is still problematic due to standard performance metrics, just on tasks beyond CIFAR-10. In other words, the importance of what you describe in point 1 seems unquantifiable to me, except with standard performance metrics.
> > > > > > 4. I very clearly am not penalizing the current paper for lack of comparison with recent works, which were not mentioned in my original assessment (my score has not changed). I brought them up to either (a) illustrate an evaluation setting that would be more convincing (you are welcome to use others such as the one of Zela at el. (2020) or (b) demonstrate using related papers what in my view constitutes a significant contribution towards search space diversity.
> > > > > > 5. I did not find the quoted line concerning computational costs on page 2 of the most recent arXiv version of NB301. My number comes from personal experience running DARTS evaluating on a V100. Note that a computational discrepancy can also be explained by differing hardware; specifically the group responsible for NB301 may very well have hardware inaccessible to other researchers.

---

> > > > > > > ### Author Response · Authors · 2021-11-30
> > > > > > > **Response**
> > > > > > >
> > > > > > >
> > > > > > > 1. Thanks for clarifying. In that case we note & respect your subjective view, but our stance on “experimentally tested specific action item” was already made clear in Final Response Parts I & II.
> > > > > > >
> > > > > > > 2. In your previous reply you refer to “significant failures in terms of error/latency metrics”, and a key reason you said why the problem we study is less interesting is because “it seems NAS algorithms successfully navigate the observed phenomena”, and that is how we inferred on what you defined as a “major problem”. By that criterion, we do not think [1] expose any “major problem” as [1] show even random search, the simplest possible method, navigates the search space quite well, with or without the problems identified.
> > > > > > >
> > > > > > > If, however, we can somewhat agree on the value of [1], especially that you acknowledge “baselines also performed well on them and they were being unduly influenced by the evaluation protocol” is a “major problem”, then we argue what we study is at least of similar importance/interest because 1) PrimSkip archs, being similar to human designed archs and sampled using extremely simple heuristics, are “baselines” and having a non-diverse search space is an “evaluation protocol”. Just like how [1] observed RS perform well and conclude there is a problem, the same argument very much applies here to show there’s “major problem” too (except our finding is even stronger: baselines do not just perform *near* SoTA, almost *any sample* performs *at* SoTA) and 2) many of our findings potentially *explain* issues in [1]: the reason for narrow performance range and that seed has a relatively large impact, especially amongst the good-performing architectures, could be due to lack of variability in architectures themselves (which we analyse and make clear) -- this is not explored in [1]. It is finally worth noting that the suggestions of [1] (Sec 6) are similarly not “empirically tested” but it doesn’t stop many of these suggestions from being useful for a follow-up work.
> > > > > > >
> > > > > > > We do not dispute your personal view on [2], but we note that the most influenced papers by [2] primarily cite their empirical results (i.e. wide cells favoured due to faster convergence). (Influence measured by [Semantic Scholar](https://www.semanticscholar.org/paper/Understanding-Architectures-Learnt-by-Cell-based-Shu-Wang/5525c7dd8dcb7f892f547804c124de457e8e9419) — apologies that we cannot just read every paper in a short timeframe so have to rely on these tools). We hope we can all agree that a paper can be of interest to the community in spite of our subjective views.
> > > > > > >
> > > > > > > 3. > but to me this seems somewhat banal
> > > > > > >
> > > > > > > Again, this is a personal view we respect (but disagree). Firstly, “banality” or “interestingness” is subjective and according to ICLR Reviewer Guidelines, should not be a ground of acceptance or rejection alone. Secondly, a large corpus of papers still cite the nominal complexity of DARTS as a justification of its purported diversity and hence benchmark on it (even if not mentioned explicitly, we argue this belief is implicit given the popularity of DARTS as an "open-domain" test space). We believe our interpretable framework and finding that DARTS architectures are actually non-diverse & similar to existing architectures due to the redundancies should not be taken as granted. Given how popular DARTS is we believe such a study merits a full paper, especially given that we do not *only* deal with that (also see Final Response Part II).
> > > > > > >
> > > > > > > > In other words, the importance of what you describe in point 1 seems unquantifiable to me, except with standard performance metrics.
> > > > > > >
> > > > > > > We believe the effect is quantifiable: 1. If the archs of different methods do not just converge to something similar, then the performance metric comparison is more meaningful. 2. The tools developed in this paper enable us not only to compare performances but also attribute them to features. If a search space is no longer non-diverse we expect novel or at least different features from the importance analysis.
> > > > > > >
> > > > > > > 4. Thanks for clarifying and for bringing these works up, which we agree are relevant and will be discussed (we cannot modify the paper after 22 Nov). However, we still argue that there isn’t a singular way to make contributions (discussed in Final Response Part I).
> > > > > > >
> > > > > > > 5.  It is present in their [v1](https://arxiv.org/pdf/2008.09777v1.pdf) (and the originally reviewed version) but we are not sure on why the authors changed that. In any case, we feel that the 1-2 GPU-hour is accurate to our experience (you can even run ~2 simultaneous training on a single V100 since the model used in NB301 is not that big). You are probably talking about training large architectures for 600 epochs to get the 5K hours figure, but as we’ve shown that is not necessary (we follow the NB301 protocol, and as we’ve shown in Sec 4 the results still generalise to the large architectures scenario but the point is you don’t have to train large architectures for every experiment).

---

### Official Review · Reviewer_7BWB · 2021-11-02

**Correctness:** 3
**Technical Novelty And Significance:** 2
**Empirical Novelty And Significance:** 3
**Recommendation:** 6
**Confidence:** 4

**Main Review:**

Pros:
- Comprehensive studies on the DARTS search space, particularly operation importance and common subgraphs. These anlysis in a sense expose that redundancy and diversity issues that exist in the cell design of DARTS.
- The paper gives some suggestions in the design of search space and benchmark for NAS: 1) search (grow the search space) and prune (remove redundancy). 2) focus on macro design choices rather than cell design choices.
- Originally I am skeptical about using surrogate benchmark, as using predicted performance may draw biased conclusion on operation importance etc. The paper did verify this as seen in Figure 5 and Figure 10 (b).

Cons:
- **Open questions are not discussed**: Despite many interesting findings, the authors left many questions in the end. What is a good search space? Can the authors suggest a list of good practices when designing a search space?
- **Mainly based on NAS-Bench-3 / DARTS search space**: The analysis is mostly based on the DARTS cell. This show that problem exists in the DARTS search space, but it doesn't suggest that cell-based search space suffer from the same problems. As the authors also mentioned in the appendix, redundancy in NAS-Bench-201 is less significant. For example, the skip constraint does not impact the performance as much as DARTS.
- **Merit of cell-based search space**: The paper heavily criticizes the cell-based search space. However, I am not convinced that the existing cell-based search space is not valuable. It is ideal if we can find SOTA model in the search space, but the main point of NAS search space is to provide a fair common ground to compare NAS methods.
- **Operation importance**: Related to the previous point, the search space provides design trade-off but the operation importance is not considering this aspect. For example, one may find a model with better FLOPS or latency. Would the 'unimportant' operations such as pooling actually contribute to reducing computation cost?


**Summary Of The Paper:**

This paper strongly criticized the current cell-based NAS approach is limitiing research in NAS. The main points are:
1. Redundancy in the cell-based search space: Performance of architecture is mostly attributed to a few important operations, while the 'unimportant' operations only add complexity to the search space. Essentially, the high-performing models share similar sub-graphs.
2. Constrianting cells to similar patterns yields better results: By limiting the search space to important operations and similar sub-graphs, random sampling in NAS performs on par with most other NAS algorithms.

**Summary Of The Review:**

The findings in this paper are inspirataional. However, the analysis is limited to particular search space which doesn't mean that the cell-based approach is not valuable in general. The paper is trying to provide some insights on search space design but I would like to see more elaboration in this direction for this paper to be accepted.

---

> ### Author Response · Authors · 2021-11-15
> **Thank you for your feedback (1/3)**
>
> We thank the reviewer for their constructive feedback! Please see our overall response and the detailed reply below, which we hope will resolve any concerns the reviewer might have; we hope that in light of this the Reviewer would consider increasing their score.
>
> >1. Open questions are not discussed... The paper is trying to provide some insights on search space design but I would like to see more elaboration in this direction for this paper to be accepted.
>
> While we do discuss some of the open problems in Sec 6, following the reviewer’s comment, we expanded on Sec 6 and added a new appendix section (App J) to elaborate more on the promising future directions. These  summarised below:
>
> - Be aware of the redundancies and do not simply augment the cell with more primitives (operation unit choices) or increase the cell sizes: we find that the current DARTS cell is already redundant with extraneous primitives. Yet, in order to get a more complicated/expressive search space, we might be tempted to add more or increase the number of nodes and connections within each cell. The findings in our paper caution against this, and we also include the tools (App F) for checking redundancies in an arbitrary new search space for future search space designers.
> - Simplify the currently redundant cell design while focusing more on searching the macro connections between cells: current cell-based NAS almost overwhelmingly focuses on cells but  manually fix the inter-cell connections in an almost linear manner similar to the classical networks (e.g. ResNet) that are known to perform well. A better option would be to simplify the structures within cells (as the current cell is already overly complicated, as we’ve shown in Sec 3/4) so as to allocate more resources (search budget) into how different cells are connected. However, most popular search spaces are either purely cell-based (DARTS) or purely macro-based (e.g. MobileNet). The cell-based structures do offer many conveniences, and we believe it should be able to find a common ground and further investigating the NAGO space in [1], which features a hierarchical, multi-level search space, could be a good starting point.
> - NAS Benchmarks beyond cells: while exceptions may exist, the most popular NAS benchmarks on computer vision problems, which greatly facilitate faster iteration of search methods, are currently based on the cell-based spaces (NAS-Bench-101/202/301/x11). An improved search space would be made more widely accepted in a much easier way if a suitable benchmarking tool is developed. Therefore, another promising direction is to build NAS benchmarks beyond the current cell-based spaces.
> - Search on lower levels (thanks to suggestions of Reviewer eT12): current search spaces  might implicitly encourage the finding of architectures  similar to the ones designed by human experts because the NAS space design often borrows heavily from the classical network designs (e.g. the macro connections between cells and the manually inserted convolution and pooling layers are often found in the manually-designed networks). While this is not necessarily bad on its own, it might be more difficult for NAS to find something novel beyond what we’ve already known. A promising direction is to go lower -- instead of searching on high-level primitives and operators, we could go search on something more fundamental like searching for sequences of mathematical operations. The pioneering work is [3], but similar to some early-stage NAS works, it was built on evolutionary algorithms, which are not particularly sample- and computation cost-efficient (as opposed to later works like one-shot/differentiable NAS which dramatically reduced computing cost of NAS). Thus, another promising direction would be to explore searches on a more fundamental level that is free of bias from existing design, but to develop more scalable and tailored solutions to make the problem more tractable
>
> *-- to be continued below--*

---

> > ### Author Response · Authors · 2021-11-15
> > **Thank you for your feedback (2/3)**
> >
> > *-- continued from above --*
> >
> > >2. Mainly based on NB301/DARTS space and is not representative of all cell-based spaces and Skip constraint does not impact the performance as much in NB201
> >
> > Firstly, NB301/DARTS space is the dominant cell-based search space in Neural Architecture Search. As we surveyed in App C, as of 2021 it is still the top (and in many cases, the only) one evaluated by the NAS papers beyond toy examples/closed-domain benchmarks. Many other cell-based spaces, such as NB201 and others discussed in Section 2, are derived, inspired or simplified from the DARTS space. Therefore, while alternative spaces may exist, we do argue that analysing the DARTS space is somewhat representative of at least the current state of the cell-based spaces in NAS.
> >
> > It is indeed true that NB201 is “less redundant”, but that is because NB201 features smaller cells with fewer nodes/edges and fewer primitive choices, and each operation naturally exerts a larger impact on the performance. Apart from that, most, if not all, key findings **do** still hold in the NB201 space: good-performing architectures can be described by simple patterns identical to what we described in DARTS space (Fig 16 vs Fig 10), and the general importance pattern of the different types of primitives (pooling, skip, conv) is again very consistent with the findings in the DARTS space (Fig 15 vs Fig 3).
> >
> > Skip just does not impact the performance as much compared to no constraints. However, noting the large and statistically significant gap (noting that the boxes do not overlap) in Prim and PrimSkip in all 3 datasets of NB201 (Fig 16), it is clear that skip connection is still essential and that replacing the residual connections on expectation leads to statistically significant performance drops. In fact, the optimal architectures for all 3 tasks, although different from each other, all satisfy both Prim and Skip constraints.
> >
> > >3. The paper heavily criticizes the cell-based search space. However, I am not convinced that the existing cell-based search space is not valuable. It is ideal if we can find SOTA model in the search space, but the main point of NAS search space is to provide a fair common ground to compare NAS methods.
> >
> > We do not claim that the cell-based designs are not valuable (we have made this clearer in Sec 6). We certainly agree that they are very valuable but also believe that there is room for improvement: cells simplify NAS significantly, and they’ve enabled cost-effective comparisons and fast iterations of the NAS methods, both of which is partly why they have become so popular. Because of its overwhelming popularity and the amount of computation/manpower committed, we argue for a healthy critical examination of its limitations, in order to make the community aware of the flaws and insufficiencies so that the designs of the search spaces can also be improved alongside the search methods. This is the motivation of our paper.  In our opinion, it is extremely unlikely that the community’s first attempt at designing an efficient search space will be optimal, and our results show that there is clearly room for improvement.
> >
> > We argue that the ultimate goal of NAS is still to find models that achieve SoTA or otherwise superior to the existing manually-designed networks in terms of accuracy, efficiency, cost, or a combination of these objectives. While we agree that it might not be a prerequisite of a benchmark to have a search space that contains these SoTA models, the benchmark should nevertheless reliably reflect or predict the performance of the different search algorithms when we would indeed like to search for the SoTA models. While current search space enables us to compare search efficiency of various NAS methods, different approach might converge to very similar architectures as we shown in the paper, making it difficult to distinguish among them based on the final performance reached and as a result, it might be difficult to make meaningful extrapolations to their relative performance in, say, an alternative, production-scale search space where we do want to search for SoTA models.
> >
> > *-- to be continued below --*

---

> > > ### Author Response · Authors · 2021-11-15
> > > **Thank you for your feedback (3/3)**
> > >
> > > *-- continued from above --*
> > >
> > > > 4. Related to the previous point, the search space provides design trade-off but the operation importance is not considering this aspect. For example, one may find a model with better FLOPS or latency. Would the 'unimportant' operations such as pooling actually contribute to reducing computation cost?
> > >
> > > We mainly focus on the architecture performance as the objective, as that is still the sole focus of the majority of the papers in the standard cell-based space (at least in the recent published papers surveyed  in App. C). While many papers report the efficiency/cost of the architectures in terms of, e.g., #params/FLOPS, they are not often  used explicitly as the search objective. A common argument is that the design space itself already implicitly contains constraints on the cost/model size (e.g. by using lower-cost operations like separable convs instead of normal ones; the number of channels for conv operations in the architectures within the DARTS space on ImageNet is capped in terms of FLOPS to accommodate for the mobile setting, etc). Furthermore, cost and performance are usually conflicting objectives; optimising multiple objectives gives rise to a Pareto front instead of a single optimum. While treatments like scalarisations of the multiple objectives are available, the trade-offs amongst the objectives remain problem specific (e.g. on a particularly low-power device like a smartwatch, we might weigh more on minimising costs. On a mobile phone, we could favour performance more). Therefore, without a clear, unequivocal objective, it would be difficult for us to conduct analysis like operation importance on these objectives (we rarely **only** care about costs; it is almost always a combination of cost and performance even when we would like to minimise costs).
> > >
> > > On the second part of the question, we find that  pareto optimal architectures, which satisfy PrimSkip constraints without using any pooling primitives, can still achieve comparable trade-off between minimising test error and minimising cost  (in terms of #params and FLOPs) compared to those with pooling primitives. The reviewer is referred to App. G in the updated paper for detailed discussions.
> > >
> > > References
> > >
> > >
> > > [1] "Neural architecture generator optimization." NeurIPS 2020.
> > >
> > > [2] "Hierarchical Representations for Efficient Architecture Search." ICLR 2018.
> > >
> > > [3] "Automl-zero: Evolving machine learning algorithms from scratch." ICML 2020.

---

> > > > ### Comment · Reviewer_7BWB · 2021-11-20
> > > > **Thanks for extra details**
> > > >
> > > > I appreciate the summary in Section 6 and Appendix J -- the suggestions of future search space design do link the findings, and make the paper solid.
> > > > The results in Appendix G also addressed my concern about the redundancy of primitives in multi-objective optimization scenario.
> > > >
> > > > Overall, I see the value of this paper and have updated my score.

---

### Official Review · Reviewer_1mk2 · 2021-11-04

**Correctness:** 3
**Technical Novelty And Significance:** 3
**Empirical Novelty And Significance:** 3
**Recommendation:** 8
**Confidence:** 4

**Main Review:**


# Merits

- The paper addresses arguably one of the main challenges in current neural architecture search, the search space design. The provided results question the popular argument that a higher dimensional search space is automatically more expressive and leads to new promising architectures.

- While one could argue that experiments are based mostly on the NASBench301 surrogate model, I do think they are sufficiently reliable. First, the surrogate models has been shown to provide reliable predictions and, second, the paper shows the same results emerge on a subset of architectures trained on the original data.

- Even though I honestly found the results of the paper less surprising, I think the paper has a high chance to impact the future development of new search spaces

- Overall I found the paper well written and easy to follow. Also, the proposed techniques to analyse operators and subgraphs seem sensible.




# Concerns

- The paper focuses mostly on the DARTS search space and NB201, which is a simplified version of the former. However, other search spaces (e.g NB101) assign operations to nodes instead of edges which might exhibit different properties than DARTS based search spaces. While I do not think that it effects the main findings, it would be great if the authors could comment on that in the paper.

- I haven't fully understood how you account for the bias towards simpler subgraphs in your subgraph-level analysis in Section 4. How does this reference set of architectures with randomly sampled operations correct this bias?


**Summary Of The Paper:**

The paper analyses cell-based search spaces for neural architecture search based on the NASBench301 dataset. The main finding are that a) only a subset of the operations actually contribute to the final performance and b) well performing architectures exhibit common patterns from the literature (e.g res-net style motifs), questioning whether we actually find entirely new configurations with current search spaces.

**Summary Of The Review:**

The paper points out several shortcomings of cell-based search spaces which are arguably the default in neural architectures search. In the long run, I think the paper can have a significant impact and help us to define better search spaces.

---

> ### Author Response · Authors · 2021-11-15
> **Thank you for your feedback**
>
> We thank the reviewer for their constructive and positive review! We refer the reviewer to the overall response and our detailed reply below, which we hope would resolve any remaining concerns the reviewer might have:
>
> > 1. The paper focuses mostly on the DARTS search space and NB201, which is a simplified version of the former. However, other search spaces (e.g NB101) assign operations to nodes instead of edges which might exhibit different properties than DARTS based search spaces. While I do not think that it effects the main findings, it would be great if the authors could comment on that in the paper.
>
> We thank the reviewer for this comment. We did not test on NB101 as it only considers CIFAR-10 like NB301, but NB301 is much larger and closer to what the community views as a “realistic” search space, and is also more amenable to one-shot/differentiable methods which are currently mainstream in NAS. We argue that the effect described in the paper should be invariant of how the architectures are represented. To give a concrete example, the DARTS/NB201 search space can similarly be represented in a feature-on-node style similar to NB101 that the reviewer described. For example, [1] uses a feature-on-node representation of architectures in an essentially enlarged version of DARTS space, and Fig 14 on Page 23 (App G1) of  [2] shows how a DARTS cell can be equivalently represented by edge-attributed and node-attributed DAGs. It is clear that these architectures, even though now node-attributed, will still be affected by the findings in the paper since we are still dealing with the identical search space.
>
> We've added discussions as the reviewer requested in Sec 2.
>
> >2. I haven't fully understood how you account for the bias towards simpler subgraphs in your subgraph-level analysis in Section 4. How does this reference set of architectures with randomly sampled operations correct this bias?
>
> We agree that due to the space constraint that this might have not been fully clear in the paper. We have updated the paper with an appendix that explains this procedure in greater detail and with some illustrations (App. I). We would like to direct the reviewer to that section but here we quote the key intuition of why the procedure corrects the bias here (also in App. I):
>
> *The key intuition here is that if a subgraph has a high support simply by virtue of its simplicity or otherwise, it should have high support values in *both* the target and null reference sets and taking the ratio would cancel out its influence (due to the high denominator value). On the other hand, if there exists an underlying mechanism such that certain subgraphs are more likely to be important (which is the primary objective of the FSM in this context -- to identify whether there exists sub-structures within cells that correlate with good performances), then they will likely be more over-represented in the target set than in the null reference.*
>
> Note that measuring significance in a similar way over some reference is also mentioned here in the Machine Learning with Graphs course by Jure Leskovec:
>
> https://www.youtube.com/watch?v=lRCDpfJoMiE&t=1324s (22:00 onward)
>
>
> References:
>
> [1] “Efficient Neural Architecture Search via Parameter Sharing,” ICML 2018.
>
> [2] “Interpretable Neural Architecture Search via Bayesian Optimisation with Weisfeiler-Lehman Kernels,” ICLR 2021.

---

> > ### Comment · Reviewer_1mk2 · 2021-11-22
> > **Response to rebuttal**
> >
> > I thank the authors for replying to my review and addressing my concerns. I will keep my score.

---

### Official Review · Reviewer_zTwP · 2021-11-04

**Correctness:** 4
**Technical Novelty And Significance:** 3
**Empirical Novelty And Significance:** 3
**Recommendation:** 6
**Confidence:** 4

**Main Review:**

Strengths:
1. The findings are very relevant to the NAS community. Rather than searching over all the operations and  spending a lot of computational effort during the search, by using all the operations, it is better to search on a subset if we know it is more promising
2. By showing that skip connection is crucial for training as a residual connection, they point out that same intuition behind the designing of ResNet.
3. Although the authors did not mention this, reducing the number of operations search space alleviates the problem of the ranking of the architectures sampled from supernet  based on their weight-sharing accuracy and the ranking of the same architectures when trained from scratch is not highly correlated. This has been plaguing the community and one of the solutions is to gradually pruning the less importations operations on an edge as the training proceeds.

Additional experiments requested:
1. Could you create an additional table by running Darts and Bananas (and other NAS algorithm of if you have more time) on PrimSkip search space for Cifar 10 and transfer it to Imagenet space and see what the best accuracy is? How much is the run time reduced by?
2. For Darts, what is the correlation of the ranking between the architectures based on their Supernet accuracy and that when trained from scratch in PrimSkip search space and the original darts search space?

Weakness:
 1. The only issue is the reliance on the surrogate model. In Figure 10(b), they did train the architectures, but 30 is very small to make a conclusive decision.  Also, for figure 10(a), could you increase the networks to 100?


**Summary Of The Paper:**

This paper examines the NASNET cell spaced search space in NASBENCH-301 (darts search space) and NASBENCH-201. They conclude that Separable convolution and skip connections are the most important operations. By replacing the current operations with just this subset and also enforcing the skip connection to be used as a residual connection, they are able to achieve accuracy very similar to the networks discovered by other SOTA algorithms. They also demonstrate that one does not have to search for reduce cell and that it can just use the same architecture as that of the normal cell. Finally, they replace the SOTA architectures found by the most popular NAS algorithms and replace the cells to use only Separable conv and skip connections and show that the accuracy of the network is very close to the original network.

**Summary Of The Review:**

This paper focuses on an important topic of understanding the search spaces better. In addition to developing new algorithms, one also needs to investigate the characteristics of the search space and how one can improve it.

---

> ### Author Response · Authors · 2021-11-15
> **Thank you for your feedback (1/2)**
>
> We thank the reviewer for their constructive and positive feedback! We are particularly grateful that the reviewer brought up aspects of the paper that could be relevant in the paper that we did not realise in the first place. Please see the overall response and the detailed reply below for our that we hope will resolve any concerns the reviewer might have; we hope that in light of this the Reviewer would consider increasing their score.
>
> >1. Could you create an additional table by running Darts and Bananas (and other NAS algorithm of if you have more time) on PrimSkip search space for Cifar 10 and transfer it to Imagenet space and see what the best accuracy is? How much is the run time reduced by?
>
> We have included the requested experiments in App. H of the paper, where we experiment on DARTS and NAS-BOWL [1] (a query-based BO method similar in spirit to BANANAS, but the authors have shown that NAS-BOWL attains clearly better performances than BANANAS). Since PrimSkip space effectively constrains the architecture sampling to a much stronger performing sub-region, we see that in both cases there is a massive speed-up compared to running these search algorithms in the original search spaces, exactly as the reviewer has pointed out (to take the example of NAS-BOWL, the best of architecture at the end of 100 iterations on the original search space is comparable to the initially sampled architecture in the constrained search space; in this sense there is at least an order-of-magnitude speedup).
>
> Unfortunately due to time and computing constraints, we are not able to obtain the full ImageNet results during the rebuttal period (ImageNet requires 8 GPUs and a few wall-clock days of training). We will endeavour to provide the requested experiments in the final version of the paper. However, it is worth noting that Table 2b shows the ImageNet results for 2 sampled architectures in the PrimSkip space, which are already on par with the state of the art. We hypothesise that effective searching should be at least as good as the results presented in Table 2b.
>
> >2. For Darts, what is the correlation of the ranking between the architectures based on their Supernet accuracy and that when trained from scratch in PrimSkip search space and the original darts search space?
>
> We thank the reviewer for suggesting that simplification of primitives might alleviate the problem of rank disorder in the supernetwork-based method (we did not realise this, so we particularly thank the reviewer for bringing this point up).
>
> However, for the PrimSkip space, we find the supernet correlation to be quite low. The reason is that PrimSkip constrains the architectures to a high-performing subregion, and as shown in Fig 10 the distribution of test accuracy within the PrimSkip group is very tight. The difficulty to rank in this case is unsurprising, as it is generally much difficult to rank the architectures correctly when their performance is highly close to each other especially when evaluation noise is further accounted for (this is also shown in Fig 1 of [2]; note the comparison between (e) and (f), and that between (g) and (h), where constraining the performances to the top 1% architectures dramatically reduces the rank correlation for all performance estimators).
>
> *-- to be continued below --*

---

> > ### Author Response · Authors · 2021-11-15
> > **Thank you for your feedback (2/2)**
> >
> > *-- continued from above--*
> >
> >
> > > 3. The only issue is the reliance on the surrogate model. In Figure 10(b), they did train the architectures, but 30 is very small to make a conclusive decision. Also, for figure 10(a), could you increase the networks to 100?
> >
> > We have increased the number of networks to 100 in Fig 10 as the reviewer requested. As for Fig 10b, we will increase the number to 50 in the final version (we haven’t been able to do that during the rebuttal as that involves a non-trivial amount of expensive training of new architectures from scratch). Nevertheless, it is worth mentioning that even though 30 is a relatively small number of samples, the trend has already been quite clear :note the boxes which indicate 25-75th percentiles do not overlap -- in fact, from 2-sample t-test, the differences between the groups are already very statistically significant even with just 30 samples: Random vs Prim p-value < 2e-7; Prim vs PrimSkip p-value < 2e-6.
> >
> > In addition to this and in further response to the comment on the reliance on the surrogate model, we do realise that purely relying on the NB301 predictions would subject the results to the biased collection of the training data in NB301.  Thus, we test all major findings **without** relying on the surrogate as well: for example, in addition to Fig 10 the reviewer mentioned, the results in Figs 5,6,8 are also from actual training of the architecture from the scratch. Similarly, Table 2 which shows the results in larger architectures on CIFAR-10 and ImageNet are again via actual training (note that these are trained even more differently compared to the NB301 pipeline; the latter only trains on CIFAR-10 with smaller 8-layer architectures). Therefore, while we rely on the surrogate model to reach the key findings, all of them are tested and verified by actually training the networks.
> >
> > References:
> >
> > [1] “Interpretable Neural Architecture Search via Bayesian Optimisation with Weisfeiler-Lehman Kernels,” ICLR 2021.
> >
> > [2] "Speedy Performance Estimation for Neural Architecture Search." NeurIPS 2021.

---

> > > ### Comment · Reviewer_zTwP · 2021-11-20
> > > **Thank you for your reply**
> > >
> > > I thank the authors for their reply. In Figure 33, The y axis label must be test error. How is the test error so low? Does that mean that DARTS and Nas-Bowl search algorithms are able to find architectures with accuracy more than 99% on Cifar10?

---

> > > > ### Author Response · Authors · 2021-11-20
> > > > **Test error is shown instead of test accuracy; it's in fraction instead of in percentage -- inconsistency fixed**
> > > >
> > > > We thank the reviewer for spotting the minor typo and inconsistency! As the reviewer rightly pointed out the y-label should be error instead of accuracy. Furthermore, the error was expressed in fraction (i.e. 0-1) not percentage (i.e. 0-100) -- so the error in the range of 5-6%, roughly in line with what we'd expect. We changed Fig 33 and multiple other places where we had the same issue to ensure that all notations are consistent in the latest updated manuscript.
> > > >
> > > > We once again thank the reviewer for pointing this out. We are happy to answer any additional questions the reviewer might have, and if the reviewer thinks that we have addressed their concerns sufficiently, we hope they could consider revising their score.

---

### Official Review · Reviewer_eT12 · 2021-11-06

**Correctness:** 4
**Technical Novelty And Significance:** 2
**Empirical Novelty And Significance:** 3
**Recommendation:** 6
**Confidence:** 4

**Main Review:**

Pros:

The paper is overall well-written and solid in terms of empirical studies. A key message to me is that high-performance architectures in the cell-based search space lie in a low-dimensional “manifold” which can be explained by several simple factors such as the inclusion of key ops and specific topological patterns. While this seems a known fact implied by several existing works on random search & performance predictors, results in this work are still quite informative (e.g., the authors explicitly summarized some of the most useful design choices of the best models).

Cons:

One of the main findings the authors reveal is that random search could perform well when augmented with simple heuristics, which further implies that the current search space is not practically interesting. This is only partially correct to me because the useful constraints discovered in the paper are arguably the outcomes of a search algorithm -- namely the one performed by nas-bench-301 (“50,000 architecture-performance pairs in the DARTS space using a combination of random sampling and more than 10 state-of-the-art yet technically diverse methods”). In other words, the constrained random search in this work is in fact heavily based on the prior knowledge discovered by search.

As far as I can tell the scope of the paper is restricted to cell-based NAS. However, people have also explored many other types of search spaces, such as the ones used in ProxylessNAS, MobileNetV3, EfficientNet, TuNAS and FBNets. Unlike the cell-based ones, those search spaces emphasize macro instead of micro decisions hence are less subject to the concerns raised. There are also other types of search spaces that are presumably less restrictive in terms of the choice of primitives, such as AutoML-Zero. While I agree with the authors that cell-based search spaces are still the mainstream for weight-sharing NAS, it would be more comprehensive to discuss the aforementioned works.



**Summary Of The Paper:**

The paper performs a detailed analysis of the DARTS search space commonly used for weight-sharing neural architecture search. The authors used several simple statistical methods to identify salient features (ops or graphlet) that can best explain the higher-quality candidates. They also looked into the discovered patterns and reached the conclusion that the popular cell-based search space is highly redundant, as simply imposing two of the discovered constraints can result in strong performance even using random search.

**Summary Of The Review:**

Overall speaking, this is a good paper attempting to address an important topic in neural architecture search regarding search space redundancies. The message is not groundbreaking in my option given several existing works with similar implications, but the analysis and results are interesting enough to be informative.

---

> ### Author Response · Authors · 2021-11-15
> **Thank you for your feedback**
>
> We thank the reviewer for their constructive and positive feedback. Please see the overall response and the detailed reply below for our  that we hope will resolve any concerns the reviewer might have; we hope that in light of this the Reviewer would consider increasing their score.
>
> > 1. One of the main findings the authors reveal is that random search could perform well when augmented with simple heuristics, which further implies that the current search space is not practically interesting. This is only partially correct to me because the useful constraints discovered in the paper are arguably the outcomes of a search algorithm -- namely the one performed by nas-bench-301 (“50,000 architecture-performance pairs in the DARTS space using a combination of random sampling and more than 10 state-of-the-art yet technically diverse methods”). In other words, the constrained random search in this work is in fact heavily based on the prior knowledge discovered by search.
>
> If the reviewer refers to the fact that we need the NB301 training set to derive the findings, the reviewer is directed to App. F where using only **200 random samples** (i.e., not using (at least not directly using) the full >50,000 NB301 samples & search algorithms mentioned) already yields similar results: e.g. we can very much use the same arguments in the main text derive the Prim and Skip constraints from Fig 31, by looking at the subgraphs there (e.g. how residual skip repeatedly appears and how s3/s5 dominate the subgraphs otherwise). This suggests that while the NB301 priors are helpful, they are not necessarily essential to discover the constraints.
>
> If the reviewer refers to the bias induced by using NB301 predictions, we agree that purely relying on them would subject the results to the bias induced by the search algorithms used by NB301, and for that reason, **for every major result we do train the architectures from the scratch**  to ensure that we are not unduly affected by this.  For example, the results in Figs 5, 6, 8, 10 are all also the actual training of the architecture from scratch. Similarly, Table 2 which shows the results in larger architectures on CIFAR-10 and ImageNet are again via actual training from scratch. Therefore, while we rely on the biased NB301 architecture data and the resultant surrogate model to reach the key findings, all of them are tested and verified by actually training the networks.
>
> Taking the example the reviewer gave that random + simple heuristics do well: while we used NB301 predictions to arrive at the formulation of these heuristics, they are used completely independently from NB301:  Fig 10b shows the distribution of the **actual** test errors & the same trend holds. Note that these architectures are sampled randomly from the entire search space with or without constraints, **not** from the training set of NB301. Thus, they are nottaffected by the biased data collection procedure and prediction surrogate of the NB301. We also show in Table 2 that the same finding holds for larger, 20-layer architectures on CIFAR-10 and ImageNet which is even more different from NB301 pipeline, which trains on CIFAR-10 only with smaller 8-layer architectures.
>
> >2. As far as I can tell the scope of the paper is restricted to cell-based NAS. However, people have also explored many other types of search spaces, such as the ones used in ProxylessNAS, MobileNetV3, EfficientNet, TuNAS and FBNets. Unlike the cell-based ones, those search spaces emphasize macro instead of micro decisions hence are less subject to the concerns raised. There are also other types of search spaces that are presumably less restrictive in terms of the choice of primitives, such as AutoML-Zero. While I agree with the authors that cell-based search spaces are still the mainstream for weight-sharing NAS, it would be more comprehensive to discuss the aforementioned works.
>
> We thank the reviewer for this comment. As the title suggests, the present paper primarily deals with cell-based NAS, which as the reviewer also acknowledges, is the current mainstream. Nonetheless, it is worth noting that the analysis tools we use are search space agnostic, although currently without benchmarks available on these alternative search spaces, the computational cost to do thorough analysis would be higher. We particularly thank the reviewer for bringing up AutoML-Zero, which represents a new paradigm that conducts search in much lower and fundamental levels that in our opinion could potentially allow us to escape from some of the dilemmas identified (e.g. how we might be implicitly biased to favour architectures similar to the known networks like ResNet due to the fixed macro-connections that already somewhat resemble the manual networks) and be a promising direction worthy of further exploration. We have added a discussion on AutoML-Zero and the other non-cell search spaces brought up by the reviewer in a paragraph in the updated paper in the conclusion.

---

> > ### Comment · Reviewer_eT12 · 2021-11-28
> > **Thank you for the reply**
> >
> > Thank you for addressing my concerns in the reply. I've read it as well as other reviews in detail, and would like to keep my original score.

---

### Author Response · Authors · 2021-11-15
**Overall response and summary of changes**

We thank all reviewers for their constructive feedback. Our paper conducts a post-hoc analysis on the popular cell-based search spaces. Via interpretability tools and cheap evaluation proxy offered by NAS benchmarks, we find a number of issues of the current cell-based space for further improvements, in particular the presence of redundancies that increase complexities but minimally improving performance, and the fact that with redundancies removed, the good architectures found can be quite non-diverse, resembling each other and to the classical networks. In light of the findings, we also offer some suggestions on potential directions to improve the current search spaces for future NAS research.

We are glad that most reviewers have appreciated the findings in our paper: “interesting and informative” (Reviewer eT12), “very relevant to the NAS community” (Reviewer zTwP), “paper can have a significant impact” (Reviewer 1mk2) and “inspirational” (Reviewer 7BWB). We hope that our response could further clarify the reviewers’ concerns and, where appropriate, we have also amended the paper accordingly. Apart from minor issues like grammar and typo, the main amended/added sections in the updated paper are highlighted in red, and here we summarise the changes. Note that for the rebuttal revision now, we have added the new contents to the end of the appendix so as to not cause confusion or inconsistency in the references to the figures and tables.

- Expanded discussions on the open questions, and suggestions on future directions (reviewer 7BWB): Section 6, and a new appendix App J where we propose and thoroughly discuss three potential new directions in light of the findings in the present paper. The suggestions of Reviewer eT12 on AutoML-Zero are also incorporated in these discussions.
- Discussions on cost of architectures (Reviewers 7BWB and 2nKR): we added a discussion in Page 5 and the details in a new appendix, App. G
- Clarifications on the frequent subgraph mining procedure (Reviewer 1mk2): we shortened the description in the main text since it was not necessarily clear, and added a new illustrated App I to describe the procedure and the intuition more thoroughly.
- Increasing number of trials on Fig 10: we increase the repetitions to 100 on Fig 10a, as requested by Reviewer zTwP
- Discussions in non-cell search spaces and AutoML in conclusion (reviewer eT12)
- Performance of search methods on the PrimSkip search space (Reviewer zTwP) in App. H.

### Revision on 20 Nov
- Minor typo and notation inconsistency in some figure captions fixed.

### Revision on 21 Nov
- Added discussion on NB101 and equivalent feature-on-node representation of architecture DAGs (reviewer 1mk2) in Sec 2.
- Misc minor issues fixed.

---

### Decision · Program_Chairs · 2022-01-20

**Decision:**

Accept (Poster)

**Comment:**

This paper makes the important, albeit somewhat unsurprising, finding, that cell-based NAS search spaces, and in particular the DARTS search space, include some operations that are much better than others. Reducing the search space to these allows even random architectures to yield good performance, similarly to the findings of "Designing Network Design Spaces", https://openaccess.thecvf.com/content_CVPR_2020/html/Radosavovic_Designing_Network_Design_Spaces_CVPR_2020_paper.html

This paper received mostly positive scores (5,6,6,8). While I agree with the negative reviewer that it would be good to study this on other benchmarks as well, I follow the positive reviewers in recommending acceptance. I encourage the authors to fix the remaining typos (there are still many) and to open source their code. This would increase the paper's impact a lot.

Finally, I would like to ask the authors to avoid protraying the misconception that we don't need large and powerful search spaces. In fact, as already hinted on in Section 6, we *do* need larger and more exciting search spaces in order to discover entirely novel architectures. Also the multi-objective nature of NAS is not to be undervalued, so the take-away of the paper should *not* be that we should design NAS benchmarks with really small & strong search spaces, but that, given a specific problem and objective, it may be prudent to evaluate whether the whole power of a given NAS search space is needed or whether it can be reduced to its essential parts.